# SPLAT REGRESSION MODELS

**Mara Daniels**
Department of Mathematics
Massachusetts Institute of Technology
`maradan@mit.edu`

**Philippe Rigollet**
Department of Mathematics
Massachusetts Institute of Technology
`rigollet@math.mit.edu`

## ABSTRACT

We introduce a highly expressive class of function approximators called Splat Regression Models. Model outputs are mixtures of heterogeneous and anisotropic bump functions, termed splats, each weighted by an output vector. The power of splat modeling lies in its ability to locally adjust the scale and direction of each splat, achieving both high interpretability and accuracy. Fitting splat models reduces to optimization over the space of mixing measures, which can be implemented using Wasserstein-Fisher-Rao gradient flows. As a byproduct, we recover the popular Gaussian Splatting methodology as a special case, providing a unified theoretical framework for this state-of-the-art technique that clearly disambiguates the inverse problem, the model, and the optimization algorithm. Through numerical experiments, we demonstrate that the resulting models and algorithms constitute a flexible and promising approach for solving diverse approximation, estimation, and inverse problems involving low-dimensional data.

## 1 INTRODUCTION

In the recent history of the deep machine learning discipline, certain problem areas have enjoyed "inflection points" wherein the attainable performance and scale have rapidly improved by many orders of magnitude. These inflection points are often directly coupled to the discovery of the *right* model architecture for the problem at hand. To name a few examples: deep convolutional networks and ResNets for image classification (Krizhevsky et al., 2012; He et al., 2016); the U-Net architecture for image segmentation and generation (Ronneberger et al., 2015; Song & Ermon, 2019); and the transformer architecture for language modeling Vaswani et al. (2017). This motivates the search for new parsimonious architectures for different problem domains, and in this work, we target low-dimensional modeling problems such as those lying at the intersection of computational science and machine learning.

We introduce a new candidate that we call the 'Splat Regression Model.' In its simplest form, the model can be written as,

$$f(x) = \sum_{i=1}^{k} v_i \mathcal{N}(x; b_i, A_i A_i^T) \qquad v_i \in \mathbb{R}^p, \quad b_i \in \mathbb{R}^d, \quad A_i \in \mathbb{R}^{d \times d}$$

where $\mathcal{N}(x; \mu, \Sigma)$ is the Gaussian density function. This rather simple architecture can be seen as a two-layer neural network with an atypical activation function, or alternatively, as a generalization of the classical Nadaraya-Watson estimator for nonparametric regression to use heterogeneous mixture weights. Toward the goal of developing a general theory for these heterogeneous mixture models, we abstract them into the form $f_\mu(x) := \mathbb{E}_{v \sim \mu_x}[v]$, where $\mu_x$ are conditionals of a probability distribution $(x, v) \sim \mu$. We develop a theoretical framework for optimizing these functions and understanding their expressivity. We further demonstrate that the splat regression model is a performant architecture for low dimensional approximation, regression, and physics-informed fitting problems, typically outperforming Kolmogorov-Arnold Network (KAN) Liu et al. (2025) and Multilayer Perceptron (MLP) models by 10-100x even with far fewer parameters.

Another major success story of large-scale machine learning is in the computer graphics literature on solving *Novel View Synthesis*, where the goal is to learn a 3D scene from 2D snapshots annotated by camera position. A major breakthrough came with the introduction of Neural Radiance Fields

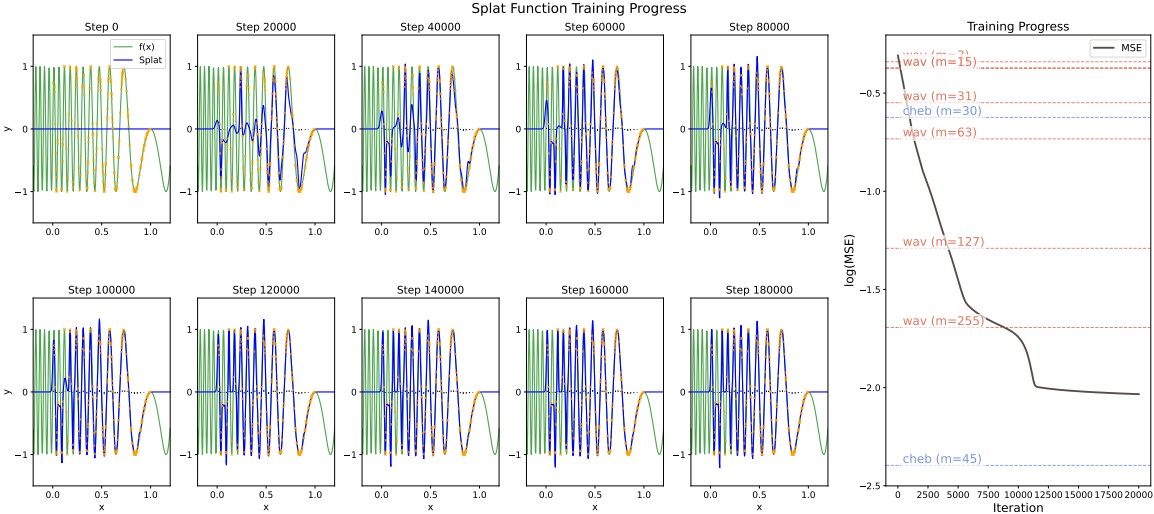

Figure 1: A representative approximation problem for the function $f^*(x) = \sin(20\pi x(2 - x))$, $d = p = 1$. We fit a $k = 30$ splat model using least squares with $n = 200$ noiseless samples and we compare to the performance of Chebyshev polynomial interpolation and Haar wavelet approximation. By learning an 'adaptive grid' interpolation, the splat regression model drastically outperforms a Haar wavelet approximations, and competes with the gold-standard Chebyshev polynomial interpolation. *(Left)*. Training iterates of a $k = 30$ splat model (blue) as it fits $f^*$ (green) by minimizing squared error with respect to $n = 200$ uniform samples (orange). *(Right)*. Validation MSE of the splat model over training. We also show the validation MSE of a Chebyshev approximation with $m = 30, 45$ gridpoints and of a Haar wavelet approximation at scales $2^l$, $l = 1, \ldots, 8$.

Mildenhall et al. (2021a). A few years later, the so-called "3D Gaussian Splatting" methodology took over as a premier architecture for graphics and reconstruction problems. There are parallels between developments in Novel View Synthesis and recent progress in physics-informed learning, which has arguably not yet enjoyed its inflection point. An essential component in NeRF modeling is the use of sinusoidal positional encoding, which drastically improves the modeling capabilities across different 'scales' of variation in the target modeling problem, and which is paralleled by the use of positional encoding schemes in physics informed PDE-fitting (Tancik et al., 2020; Zeng et al., 2024), to moderate success. In both cases, rendering solutions by pointwise evaluation of an MLP across the spatial domain can be prohibitively slow. Recognizing that Novel View Synthesis is inherently a spatial inverse problem, **a major goal of our work is to replicate the successes of 3D Gaussian Splatting across a wide variety of physical modeling and inverse problems**. We intend for this work to provide an instruction manual for deploying and training splat regression models in these settings. Our main contributions are the following.

1. We introduce the Splat Regression Model. We prove some preliminary structural theorems about this model and we show that it is a universal approximator.

2. We develop principled optimization algorithms for gradient-based training, which are crucial to our empirical success. To do this, we leverage an interpretation of the splat model parameters as a hierarchical 'distribution over distributions,' and we invoke the theory of Wasserstein-Fisher-Rao gradient flows to compute gradient updates in parameter space.

3. We recover 3D Gaussian Splatting as an instance of splat regression modeling. In Example 2, we detail a clean formulation of it, where different aspects of the pipeline are split into clear, 'modular' parts.

4. We test the performance of splat modeling in a few representative modeling problems including low-dimensional regression (Figure 2) and physics-informed fitting.

## 2 RELATED WORK

**Novel View Synthesis**. The Novel View Synthesis in computer graphics has a long history. Input data is typically sourced using Structure From Motion (SfM) methods (Ullman, 1979; Özyeşil et al., 2017). In the idealized setting, scenes are rendered using the Radiative Transfer Equation (Chandrasekhar, 2013). Much work has gone into developing fast approximations to the RTE, such as the widely used $\alpha$-blending technique introduced in (Porter & Duff, 1984; Carpenter, 1984). For 3D scene representation, Zwicker et al. (2002b) introduced the "elliptical weighted average" method, although these ideas were not applied at scale for about two decades (Kerbl et al., 2023a). Major interest in this problem was sparked with the release of Neural Radiance Fields Mildenhall et al. (2021b), an approach based on optimizing MLP parameters by differentiating through (an approximation of) the RTE.

**Physics-informed Modeling**. Following the introduction of PINNs (Raissi et al., 2019), much work has gone into developing ML methods for solving computational science problems. As documented by Krishnapriyan et al. (2021), PINN training can have surprising failure modes and requires careful tuning and hand-crafted architectures. More recently, adding positional encoding schemes Tancik et al. (2020); Huang et al. (2021) through either RBF or sinusoidal encoding was shown to improve performance. Alternatively, the Kolmogorov-Arnold Network architecture Liu et al. (2025); Rigas & Papachristou (2025) was introduced as a competitor, and some comparisons between KAN and RBF interpolation methods are explored in Li (2024). One way to view splat regression modeling is as learning an 'adaptive interpolation grid,' and our experimental results suggest that 'smart positional encoding is all you need' for low-dimensional modeling problems.

**Mean-field Theory of Two-layer Networks**. Wasserstein-Fisher-Rao gradient flows can be used to study the optimization of two-layer ReLU neural networks. This approach was pioneered by Chizat & Bach (2018), who prove a conditional global convergence result for two-layer network optimization. Further work in this direction includes (Mei et al., 2019a;b). The main difference between optimizing ReLU networks and Splat models is that our model has a fundamentally different geometry of its parameter space, since first-layer output neurons are represented as elements of a Bures-Wasserstein manifold. This is also a major difference between our work and Chewi et al. (2025b), which introduces a modified ReLU network architecture whose neurons can be 'superpositions' of ReLU functions.

## 3 SPLAT REGRESSION MODELS

We now state a few essential definitions and basic properties of splat regression models. Readers who are uninterested in the abstract formulation may skip to equation 1, which is the concrete object that appears in algorithms. In Example 1, we discuss how to fit these models by minimizing loss on a training dataset.

We write $\mathcal{L}_\pi^s(\mathbb{R}^d; \mathbb{R}^p)$ to denote the functions $f : \mathbb{R}^d \to \mathbb{R}^p$ for which $\int \|f(x)\|_s^s \, \pi(dx) < \infty$, where $\pi$ is a measure on $\mathbb{R}^d$. When $s = 2$, we endow it with the inner product $\langle f, g \rangle_{\mathcal{L}_\pi^2} = \int \langle f(x), g(x) \rangle \, \pi(dx)$. We write $C_b^s(\mathbb{R}^d)$ to denote the real-valued $s$-times continuously differentiable functions on $\mathbb{R}^d$ with $s$ bounded derivatives. For a function $\phi : \mathbb{R}^d \to \mathbb{R}$, we write $\nabla \phi(x)$ its gradient, $\nabla^2 \phi(x)$ its Hessian, and $\Delta \phi(x) = \sum_{i=1}^d \partial_{x_i}^2 \phi(x)$ its Laplacian. For $\psi : \mathbb{R}^d \to \mathbb{R}^p$ we write $D\psi(x)$ its Jacobian and (when $p = d$) we write $\operatorname{div} \psi(x) = \sum_{i=1}^d \partial_{x_i} \psi^{(i)}(x)$ its divergence. We denote by $\mathcal{P}(\Omega)$ the space of probability distributions on $\Omega$, by $\mathcal{P}_s(\Omega)$ the distributions with $s \geq 1$ finite moments, and by $\mathcal{P}_{ac}(\Omega)$ the distributions with a density. For $\mu \in \mathcal{P}(\Omega)$ we write $\mathbb{E}_{X \sim \mu}[f(x)] = \int f(x) \, \mu(dx)$ and, if $\mu$ has a density, we denote it as $\mu(x) : \Omega \to \mathbb{R}$. Last, we denote by $T_{\#}\mu$ the pushforward of $\mu$, that is, the distribution of $T(X)$ when $X \sim \mu$. We write $\operatorname{id}(x) = x$ the identity map, $\operatorname{id}_{\#}\mu = \mu$.

We begin by defining the family of *splat models* that we consider in this work. We first define the family of splats and then explain how they are used in splat modeling.

**Proposition 1** (Splats are a generalized Bures-Wasserstein manifold). *Let $\rho \in \mathcal{P}(\mathbb{R}^d)$ be a centered isotropic mother splat. We denote the set of all splats as,*

$$\mathsf{BW}_\rho(\mathbb{R}^d) := \left\{ (A(\cdot) + b)_{\#}\, \rho : A \in \mathbb{R}^{d \times d}, b \in \mathbb{R}^d \right\}.$$

*Then* $\mathsf{BW}_\rho(\mathbb{R}^d)$ *is a geodesically convex subset of* $\mathcal{W}_2(\mathbb{R}^d)$*, and on this space the Wasserstein metric reduces to the Bures-Wasserstein metric (Modin, 2016; Bhatia et al., 2019),*

$$W_2^2(\rho_{A,b}, \rho_{R,s}) = \|b - s\|_2^2 + \|A\|_F^2 + \|R\|_F^2 - 2\|A^T R\|_* \qquad A, R \in \mathbb{R}^{d \times d} \quad b, s \in \mathbb{R}^d$$

*where* $\| \cdot \|_F$ *is the Frobenius norm and* $\| \cdot \|_*$ *is the nuclear norm.*

We give a short, self-contained proof of this proposition in Appendix B.2. We occasionally write $\rho_{A,b} := (A(\cdot) + b)_\# \rho$ when we wish to call attention to the parameters $(A, b)$ of the splat. These parameters can be thought of as a global coordinatization of $\mathsf{BW}_\rho(\mathbb{R}^d)$. We discuss further the Wasserstein distance and its associated geometry in Section 3.2.

We envision $\rho \in \mathsf{BW}_\rho(\mathbb{R}^d)$ as an isotropic 'bump' function and call each individual copy a *splat* in homage to the computer graphics literature where an instance of this model was first widely popularized. We call $\rho$ the *mother splat* in analogy to the *mother wavelets* used in wavelet theory (Mallat, 2008). Splat models are defined as follows.

**Definition 1** (Splat Measures and Splat Models). Let $\rho \in \mathcal{P}_{ac}(\mathbb{R}^d)$ have zero mean and identity covariance, and let $\rho_{A,b}$ be the distribution of $AX + b$ when $X \sim \rho$. We say that $\mu \in \mathcal{P}(\mathbb{R}^p \times \mathsf{BW}_\rho(\mathbb{R}^d))$ is a *splat measure* and the associated splat model $f_\mu(x)$ is,

$$f_\mu(x) := \mathbb{E}[v\rho_{A,b}(x)] \qquad v, \rho_{A,b} \sim \mu.$$

If the support of $\mu$ has $k < \infty$ elements, we say $\mu$ is a *k-splat measure*, $f_\mu$ is a *k-splat model*, and we write

$$f_\mu(x) = \sum_{i=1}^{k} v_i \, \rho_{A_i, b_i}(x) \, |\det A_i^{-1}| \, \mu(v_i, \rho_{A_i, b_i}) \qquad \{(v_i, \rho_{A_i, b_i})\}_{i=1}^{k} = \operatorname{supp} \mu. \tag{1}$$

$$= \sum_{i=1}^{k} v_i \, \rho(A_i^{-1}(x - b_i)) \, |\det A_i^{-1}| \, \mu(v_i, \rho_{A_i, b_i})$$

We write $\mathcal{S}_{p,d} := \mathcal{P}(\mathbb{R}^p \times \mathsf{BW}_\rho(\mathbb{R}^d))$ (resp. $\mathcal{S}_{p,d}^{(k)}$) to denote the set of splat measures (resp. $k$-splat measures).

As we later discuss, $\mathcal{S}_{p,d}$ may be endowed with a natural metric structure, allowing us to perform gradient-based training of $k$-splat models.

### 3.1 REGULARITY AND UNIVERSAL APPROXIMATION

We note the following basic condition for $f_\mu(x)$ to be continuously differentiable.

**Proposition 2** (Sufficient conditions for regularity). *Let* $\mu \in \mathcal{S}_{p,d}$ *and suppose that the mother splat* $\rho$ *has* $s \geq 0$ *bounded derivatives. Then* $f_\mu(x)$ *also has* $s$ *bounded derivatives.*

It is also helpful to understand the expressiveness of the class of splat models, and in particular, to understand what kinds of functions can be approximated by $k$-splat models for finite $k$. As we show in Appendix A, the finite splat model equation 1 is already in the scope of Cybenko's early result on universal approximation of real-valued functions by two-layer neural networks.

**Proposition 3** (Corollary of Cybenko (1989), Definition 1 and Theorem 1). *Let* $\Omega$ *be a compact subset of* $\mathbb{R}^d$ *and suppose* $\rho$ *is a continuous density with marginal* $\rho_1(x_1) = \int_{\mathbb{R}^{d-1}} \rho(x_1, x_2 \ldots x_d) \, dx_2 \ldots dx_d$. *Then for any* $f \in C_b^0(\Omega; \mathbb{R})$ *and* $\epsilon > 0$*, there exists a* $k$-splat *measure* $\mu$ *(with mother splat* $\rho$*) such that*

$$\sup_{x \in \Omega} |f(x) - f_\mu(x)| < \epsilon.$$

We also prove in Appendix A a quantitative bound on the number of splats required to approximate any Lipschitz function on a 'nice' domain using any 'nice' mother splat.

**Theorem 3** (Informal – quantitative universal approximation) **.** *For any bounded, 'nice'* $\Omega \subseteq \mathbb{R}^d$*, any bounded, Lipschitz* $f : \Omega \to \mathbb{R}^p$*, and any bounded, Lipschitz mother splat with finite first moment, there exists for each* $\epsilon > 0$ *a* $k$-splat *measure* $\mu$ *with* $k \lesssim \epsilon^{-2(d+2)}$ *and,*

$$\sup_{x \in \Omega} \|f_\mu(x) - f(x)\|_2 < \epsilon.$$

To complement this, we give the following lower bound on the error of uniform approximation over the family of 1-Lipschitz functions, whose proof is also contained in Appendix A.

**Theorem 4** (Informal – universal approximation lower bound) **.** Let $\rho$ be an $L_\rho$-lipschitz, $B_\rho$-bounded mother splat density. Fix $k \geq 1$ and $h > 0$, and set

$$\mathcal{S}_h := \left\{ f := \sum_{i=1}^{k} v_i (A_i(\cdot) + b_i)_{\#} \rho \ : \ (v_i, A_i, b_i) \in \Gamma, \ i = 1 \ldots k \right\}$$

$$\mathcal{F} := \left\{ f : [0,1]^d \to \mathbb{R} : f(0) = 0, \ f \text{ is 1-Lipschitz} \right\}$$

where $\Gamma$ contains all splat parameters $(v, A, b) \in \mathbb{R} \times \mathbb{R}^{d \times d} \times \mathbb{R}^d$ with $v, b$ bounded and $\sigma_{\min}(A) \geq h$, where $\sigma_{\min}(A)$ is the smallest singular value of $A$. If it holds that,

$$\sup_{f \in \mathcal{F}} \inf_{\hat{f} \in \mathcal{S}_h} \|f - \hat{f}\|_\infty \leq \epsilon \tag{2}$$

then (up to logarithmic factors) $\epsilon^{-d} \lesssim k d^2$.

The lower bound rate is essentially tight up to logarithmic factors: any $f \in \mathcal{F}$ admits a piecewise constant approximation on sets of the form $\bigotimes_{i=1}^{d} [s_i \epsilon, (s_i + 1)\epsilon)$ where each $(s_i)_{i=1\ldots d}$ has $1 \leq s_i \leq d$ and where $\epsilon = k^{-1/d}$. Such an approximation is representable by a $k = \epsilon^{-d}$ splat model by taking mother splat $\rho = [-c, c)^d$ for an appropriate $c > 0$. Taken together, Theorem 3 and Theorem 4 imply that there exists a (non-Lipschitz) mother splat that achieves the minimax optimal rate $\epsilon \sim k^{-1/d}$ of approximation of Lipschitz functions, and also, that for any 'nice' mother splat, the approximation rate is at worst $\epsilon \sim k^{-1/2(d+2)}$. However, much like the corresponding universal approximation rates for multilayer perceptrons, these worst case approximation bounds typically do not describe the number of parameters required to achieve a high quality fit for real data. We discuss further in Section 4.

### 3.2 GEOMETRY OF SPLAT MODELS

Ignoring some technical caveats, the space $\mathcal{P}(\mathbb{R}^d)$ of probability distributions can be viewed as a manifold by assigning it a distance metric such as $W_2(\mu, \nu)$, the 2-Wasserstein distance, or $H(\mu, \nu)$, the Hellinger distance. In both cases the distance induces a metric in the geometric sense, and a tangent space structure, which together can be used to construct gradient-descent-like algorithms over the space of probability distributions. In Wasserstein space, moving between points $\mu_0, \mu_1 \in \mathcal{P}_2(\mathbb{R}^d)$ is accomplished by applying "infinitesimal" transport maps and the geodesics follow the straight line interpolations of optimal transport maps, $\mu_t = (t \, T_{\mu_0 \to \mu_1} + (1 - t) \, \mathrm{id})_{\#} \mu_0$. Under the geometry of the Hellinger metric, known as the Fisher-Rao or Information geometry Amari (1983), moving between points is accomplished by "mass teleportation," which means directly scaling the density up or down, pointwise over the domain, in a mass-preserving way. Its geodesics are $\mu_t(x) = (\alpha_t \sqrt{\mu_0}(x) + \beta_t \sqrt{\mu_1}(x))^2$, where $\alpha_t, \beta_t$ are spherical linear interpolation coordinates, as can be seen from the $\mathcal{L}^2$ form of $H^2(\mu_0, \mu_1) = \int (\sqrt{\mu_0}(x) - \sqrt{\mu_1}(x))^2 \, dx$ for densities $\mu_0, \mu_1 \in \mathcal{P}_{ac}(\mathbb{R}^d)$. Finally, it is also possible to consider the geometry that arises when allowing movements by transport and teleportation at the same time, which is the *Wasserstein-Fisher-Rao geometry* (Chewi et al., 2025a).

The Wasserstein and Fisher-Rao geometric perspectives are widely used in the design and analysis of probabilistic algorithms, such as for dynamical sampling (Chewi et al., 2025a), variational inference (Lambert et al., 2022), barycentric interpolation (Gouic et al., 2019; Chewi et al., 2020), and lineage tracing (Schiebinger et al., 2019). We apply both to design a principled gradient-based training method for splat regression. As a byproduct, we recover the heuristic methods used to optimize Gaussian Splat models, which has a clean interpretation as regularized risk minimization via Wasserstein-Fisher-Rao gradient descent. We first explain the Wasserstein structure of the space of splats, then the Wasserstein-Fisher-Rao structure of the space of splat measures.

Relative to particle methods, lifting to $\mathcal{S}_{p,d}$ allows one to implement 'smoothed particles' on $\mathcal{P}(\mathbb{R}^p \times \mathbb{R}^d)$, thus providing a computationally tractable way to flow measures that are everywhere positive and absolutely continuous. We comment in Appendix B on the relationship between gradient flows in $\mathcal{S}_{p,d}$ and in $\mathcal{P}(\mathbb{R}^p \times \mathbb{R}^d)$. This generalizes the concept of 'Gaussian particles' introduced in

(Lambert et al., 2022), because particles are represented by any affine pushforward of an arbitrary density $\rho$. Such a particle representation was also hinted at in the computer graphics literature on 'volume splatting' (Zwicker et al., 2002a), and has also appeared in early work on hydrodynamics simulations (Gingold & Monaghan, 1977).

We endow $\mathcal{S}_{p,d}$ with the Wasserstein and/or Fisher-Rao metric, similar to the 'Wasserstein over Wasserstein' approach taken in (Lambert et al., 2022; Bonet et al., 2025). After stating our results, we give Examples 1, 2 that illustrate these objects concretely in two practical settings.

**Definition 2** (First variation). Let $\mathcal{F} : \mathcal{L}^2(\mathbb{R}^d; \mathbb{R}^p) \to \mathbb{R}$ be a functional. Its first variation (when it exists) is the function $\delta\mathcal{F}[f](x)$ which satisfies for every $f, \xi \in \mathcal{L}^2(\mathbb{R}^d; \mathbb{R}^p)$,

$$\partial_\epsilon \mathcal{F}(f + \epsilon\xi) \mid_{\epsilon=0} = \int_{\mathbb{R}^d} \langle \xi(x), \delta\mathcal{F}[f](x) \rangle \, dx.$$

It is convenient to express the Wasserstein gradient $\mathbb{W}_\mu \mathcal{F}(f_\mu) : \mathbb{R}^p \times \mathsf{BW}_\rho(\mathbb{R}^d) \to \mathbb{R}^p \times \mathsf{BW}_\rho(\mathbb{R}^d)$ in the global coordinate system $(v, A, b) \cong (v, \rho_{A,b})$. This coordinate system is convenient for writing the gradient $\mathbb{W}_\mu \mathcal{F}(f_\mu) = (\mathbb{W}_v, \mathbb{W}_A, \mathbb{W}_b)\mathcal{F}(f_\mu)$, so that the particle dynamics

$$\dot{v}_t = -\mathbb{W}_v \mathcal{F}(f_\mu)(v, A, b) \qquad \dot{A}_t = -\mathbb{W}_A \mathcal{F}(f_\mu)(v, A, b) \qquad \dot{b}_t = -\mathbb{W}_b \mathcal{F}(f_\mu)(v, A, b).$$

implement a Wasserstein gradient flow in $\mathcal{S}_{p,d}$. Our main theorem is the following.

**Theorem 1** (Informal – Wasserstein-Fisher-Rao gradient of $\mu \mapsto \mathcal{F}(f_\mu)$). *Let $\mu \in \mathcal{S}_{p,d} := \mathcal{P}(\mathbb{R}^p \times \mathsf{BW}_\rho(\mathbb{R}^d))$ and let $\mathcal{F} : \mathcal{L}^2(\mathbb{R}^d; \mathbb{R}^p) \to \mathbb{R}$ be a functional. Assume that $\rho$ has a sub-exponential density. Then the Fisher-Rao gradient is given by,*

$$\nabla_\mu^{\mathsf{FR}} \mathcal{F}(f_\mu)(v, A, b) = \mathbb{E}_{X \sim \rho_{A,b}}[\langle \delta\mathcal{F}(X), v \rangle] - \mathbb{E}_{v,A,b \sim \mu}[\mathbb{E}_{X \sim \rho_{A,b}}[\langle \delta\mathcal{F}(X), v \rangle].$$

*and the Wasserstein gradient is given by,*

$$\mathbb{W}_v \mathcal{F}(f_\mu)(v, A, b) = \mathbb{E}_{X \sim \rho_{A,b}}[\delta\mathcal{F}(X)]$$
$$\mathbb{W}_A \mathcal{F}(f_\mu)(v, A, b) = -\mathbb{E}_{X \sim \rho_{A,b}} \left[ \langle \delta\mathcal{F}(X), v \rangle \left( I_d + \nabla_x \log \rho_{A,b}(X)(X - b)^T \right) A^{-T} \right]$$
$$\mathbb{W}_b \mathcal{F}(f_\mu)(v, A, b) = -\mathbb{E}_{X \sim \rho_{A,b}}[\langle \delta\mathcal{F}(X), v \rangle \nabla_x \log \rho_{A,b}(X)]$$

*where the argument of $\delta\mathcal{F}(\cdot) = \delta\mathcal{F}[f](\cdot)$ was suppressed.*

The condition on $\rho$ is required to apply integration by parts without a boundary term, but when this does not hold (such as when $\rho(x) = \mathbf{1}\{\|x\|_2 \leq 1\}$) one can derive the appropriate corrections on a case-by-case basis. We conclude with a few concrete examples of $\mathcal{F}$ which are relevant to applications.

**Example 1** (Empirical risk minimization). Suppose $x_1, x_2, \ldots, x_n \sim \mathcal{U}(\Omega)$ are i.i.d. samples, where $\Omega = [0, 1]^d$ for simplicity, and set $y_i = f^*(x_i)$ with

$$\mathcal{F}(f) := \frac{1}{n} \sum_{i=1}^n L(f(x_i), y_i)$$

for a loss function $L(\hat{y}, y)$. To calculate the variation

$$\partial_{\epsilon=0} \mathcal{F}(f + \epsilon\xi) = \partial_{\epsilon=0} \left\{ \frac{1}{n} \sum_{i=1}^n L(f(x_i) + \epsilon\xi(x_i), x_i) \right\} = \frac{1}{n} \sum_{i=1}^n \langle \xi(x_i), \nabla_{\hat{y}} L(f(x_i), x_i) \rangle$$

and so $\delta\mathcal{F}[f](x_i) = \frac{2}{n} \sum_{i=1}^n \nabla_{\hat{y}} L(f(x_i), x_i)$, which is otherwise undefined for $x \in \mathbb{R}^d$ outside the sample. We can use importance sampling to estimate the gradients,

$$\mathbb{E}_{X \sim \rho_{A,b}}[\delta\mathcal{F}[f](X)(\ldots)] \approx \frac{1}{n} \sum_{i=1}^n \rho_{A,b}(x_i) \delta\mathcal{F}[f](x_i)(\ldots).$$

This estimator is unbiased because $\{x_i\}_{i=1}^n \sim \mathcal{U}([0, 1]^d)$. For samples drawn from an unknown distribution, one could instead estimate $\delta\mathcal{F}[f](x)$ by interpolating the available data, then approximating the average over $X \sim \rho_{A,b}$ using either Monte-Carlo or a quadrature rule.

**Example 2** (Inverse problems and physics-informed training). We can also take

$$\mathcal{F}(f) = \frac{1}{2}\|\mathcal{A}[f](x) - g(x)\|_{\mathcal{L}^2(\Omega)}^2 \tag{3}$$

where $\mathcal{A}$ is a known integro-differential operator, $g$ is a known forcing function, and we again take $\Omega = [0,1]^d$ for simplicity. To calculate the variation,

$$\begin{aligned}
\partial_{\epsilon=0}\mathcal{F}(f + \epsilon\xi) &= \partial_{\epsilon=0}\left\{\frac{1}{2}\int \|\mathcal{A}[f + \epsilon\xi](x) - g(x)\|_2^2\right\} \\
&= \int \langle \mathcal{A}[f](x) - g(x), D_f\mathcal{A}[\xi](x)\rangle\, dx \\
&= \int \langle (D_f\mathcal{A})^*\mathcal{A}[f](x) - (D_f\mathcal{A})^*[g](x), \xi(x)\rangle\, dx.
\end{aligned}$$

where $D_f\mathcal{A}$ is the linearization of $\mathcal{A}$ at $f$ and $(D_f\mathcal{A})^*$ is its adjoint. So $\delta\mathcal{F}[f] = (D_f\mathcal{A})^*\mathcal{A}[f](x) - \mathcal{A}^*[g](x)$. Two illustrative instances of this setup are,

1. *A 'Physics-informed' splat solver for the Poisson equation.* Take $p = 1$, $\mathcal{A}[f_\mu] = \Delta f_\mu$. The Fisher-Rao gradient is,

$$\nabla_\mu^{\mathsf{FR}}\mathcal{F}(f_\mu)(v, A, b) = \int_\Omega v(\Delta f(x) - g(x))\,\Delta\rho_{A,b}(x)\,dx.$$

   In the same way, the coordinates of the Wasserstein gradient can be expressed as integrals with respect to $\Delta\rho(x)$ and $\nabla(\Delta\rho)(x)$. For simple $\rho$, these functions can be precomputed to accelerate gradient computations, and the integral approximated by Monte-Carlo.

2. *Splat regression modeling for Novel View Synthesis (NVS).* NVS can be viewed as an inverse problem whose forward operator $\mathcal{A}$ is the *Radiative Transfer Equation* (RTE) (Chandrasekhar, 2013), (Zwicker et al., 2002a, Equation (7)), whose unknown parameters are: the 'emission function,' $s : \mathbb{R}^3 \times S^2 \to \mathbb{R}$, and the 'extinction function' $\sigma : \mathbb{R}^3 \to \mathbb{R}$. The emission $s(x, v)$ is the amount of light radiating from $x \in \mathbb{R}^d$ outwards in direction $v$, and the extinction $\sigma(x)$ represents the degree to which point $x$ 'occludes' points behind it. Following (Kerbl et al., 2023a), we parametrize splat models $\sigma(x) = g_\nu(x)$ and we parameterize $s(x, v)$ as,

$$s(x, \cdot) = \sum_{i=1}^p f_\mu^{(i)}(x)\phi_i(v)$$

   where $\{\phi_i\}_{i=1}^p$ are spherical harmonic basis functions and $f_\mu : \mathbb{R}^3 \to \mathbb{R}^p$ is another splat model ($p \approx 20$). Given these two fields, rendering the scene involves evaluating the *Radiative Transfer Equation*.

$$\mathcal{A}[s, \sigma](x, v) = \int_0^\infty s(x + tv, v)\sigma(x + tv)\exp\left(-\int_0^t \sigma(x + sv)\,ds\right)dt. \tag{4}$$

   The RTE is typically evaluated using a discrete approximation called $\alpha$-blending (Zwicker et al., 2002b; Kerbl et al., 2023a) and splat parameters are trained via SGD with particle birth-death dynamics to minimize residuals (equation 3).

Finally, we remark that the performance of Gaussian splatting in large-scale Novel View Synthesis depends heavily on the use of many training heuristics, such as initialization of splat locations Kerbl et al. (2023b), selective noising of splat locations during training Kheradmand et al. (2024), and on 'pruning strategies' to move spurious splats either by deleting them or teleporting them elsewhere in space (Kerbl et al., 2023b; Hanson et al., 2025). By providing a principled optimization perspective on fitting splat models, our results pave the way to interpret these heuristics through the lens of *regularized risk minimization*. For instance, in a sampling context, the stochastic *Unadjusted Langevin Algorithm* can be interpreted as Wasserstein gradient descent of the functional $\mathcal{F}(\mu) = D_{\mathrm{KL}}(\mu \mid \mu_{\mathrm{target}})$ divergence, which enjoys strong convexity whenever $\mu_{\mathrm{target}}$ is strongly log concave. We anticipate that the selective noising heuristic introduced by Kheradmand et al. (2024) can be interpreted as adding a convex entropic regularizer to the objective (3), improving convergence. Similarly, particle birth-death dynamics are a well-known discretization algorithm for Fisher-Rao gradient flows Lu et al. (2023). With an appropriate discretization, Theorem 1 thus prescribes a pruning criterion that is guaranteed (in the continuous-time limit) to decrease the loss. While it is outside the scope of the present manuscript, principled regularization methods for splats is a very promising direction for future work.

## 4 EXPERIMENTS

In this section, we demonstrate the power of the splat regression modeling by directly comparing it on approximation, regression, and physics-informed modeling problems.

### 4.1 MULTISCALE APPROXIMATION

First, we show in Figure 1 a representative 1D example of training a splat regression model, which we compare to two standard function approximation algorithms: *chebyshev polynomial interpolation* and *wavelet basis projection*.

In the left subpanel, we show snapshots of splat training arranged from right to left, top to bottom. We train a $k = 30$ splat model initialized with $v_i = 0, b_i = (i - 1)/k$, and $A_i = 1/2k$ for $i = 1 \ldots k$. Training is performed by least squares fitting via Wasserstein gradient descent with learning rate $10^{-4}$ and no momentum. The loss function is $\mathcal{L}(f_\mu) = \frac{1}{n} \sum_{i=1}^{n} (f(x_i) - y_i)^2$ for $n = 200$ noiseless samples $x_i \sim \mathcal{U}([0, 1])$. In the right subpanel, we show the *validation log-MSE* of the splat model. Anecdotally, the exponentially fast convergence which appears in Figure 1 (right) is robust to different initializations and functions. We show in Appendix C, Figure 5 a version of this experiment where the $\{b_i\}_{i=1}^{k}$ are initialized as a $k$-point Chebyshev grid as well as a version where data is sampled from a discontinuous sawtooth function.

We select this example as a simple one dimensional picture of how splat models are well suited to fit 'multi-scale' features of the observed data. For comparison, horizontal lines indicate the validation MSE of the Chebyshev and wavelet approximations. The splat model with $k = 30$ splats significantly outperforms the Chebyshev interpolation with $m = 30$ interpolation nodes. Setting $m = 45$ in order to control for the total number of model parameters ($(3 \times k = 90)$ for splat vs. $(2 \times m = 90)$ for Chebyshev), we see that the splat model achieves slightly worse approximation error. Relative to wavelet approximation, the splat model outperforms a Haar wavelet projection with $m = 255$ parameters (corresponding to level $l = 8$ of the basis hierarchy). We find these results extremely encouraging, particularly given that very simple optimization and initialization schemes are enough to successfully train the splat model.

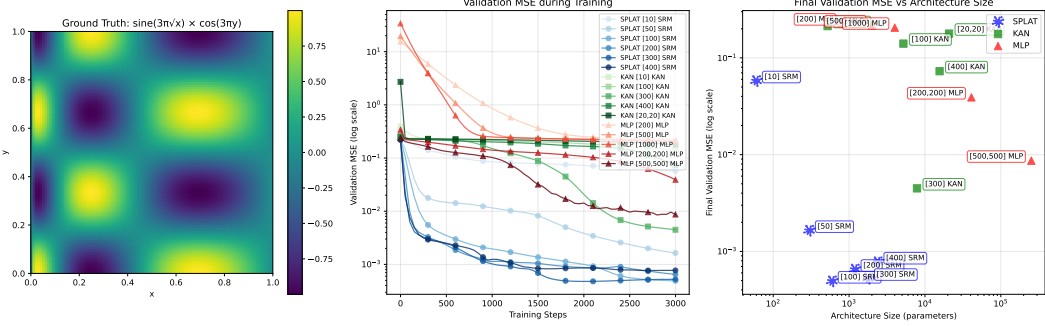

Figure 2: We compare splat regression, Kolmogorov-Arnold Networks Liu et al. (2025), and fully connected Multi-layer Perceptron in a noisy regression task. We observe that splat models achieve order of magnitude lower fitting error while using a small fraction of the parameters of MLP and KAN networks.

### 4.2 REGRESSION WITH KAN, MLP, AND SRM

We compare the performance of splat regression modeling to Kolmogorov-Arnold networks Liu et al. (2025) and to fully connected MLP architectures on a 2D regression task. We sample $x_i \sim \mathcal{U}([0, 1]^2)$ i.i.d. and set $y_i = f(x_i) + \epsilon_i$ where $\epsilon_i \sim \mathcal{N}(0, 0.01)$ and,

$$f(x, y) = \sin(3\pi\sqrt{x}) \cos(3\pi y).$$

Each model is trained using 3000 iterations using Adam with learning rate $10^{-4}$.

We attribute the improved performance of splat models to their spatially localized nature, which can be viewed as a learned positional encoding scheme. However, due to their expressivity, splat models may be more susceptible to overfitting, requiring regularization to achieve good fits.

## 4.3 PHYSICS-INFORMED MODELING

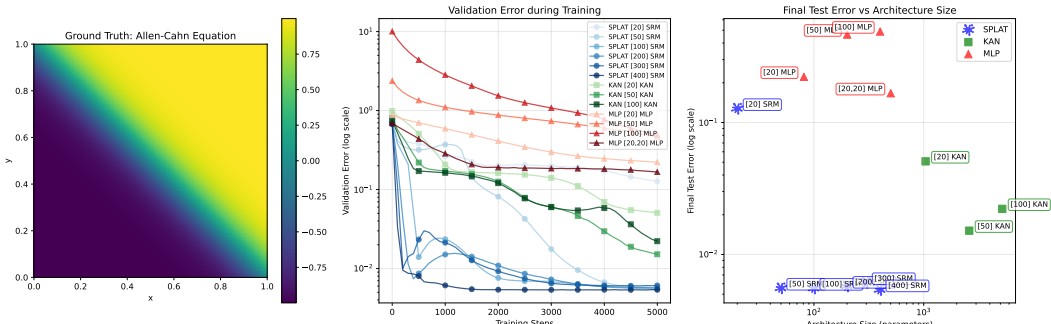

Figure 3: We compare splat regression, Kolmogorov-Arnold Networks Liu et al. (2025), and fully connected Multi-layer Perceptron in a physics informed regression task. Models are fit to solve the Allen-Cahn equation on $[0, 1]^2$. *(Left)*. True solution under this parameter regime. *(Middle)*. Validation error for each model class as a function of the number of training iterations. *(Right)*. Validation error relative to total number of model parameters. Among the test pool, a $k = 50$ splat model outperforms all KAN and MLP architectures by an order of magnitude while using significantly fewer parameters.

We further test the splat regression model in a two dimensional physics-informed learning problem where the goal is to estimate the solution of the *Allen Cahn* equation,

$$\epsilon^2 \Delta u(x) + u(x) - u^3(x) = 0 \qquad x \in [0, 1]^d.$$

This equation is a well known example of a PDE whose solutions can have 'boundary interfaces' that converge to discontinuities as $\epsilon \to 0$. We take $\epsilon = 0.1$ and train each model to minimize a weighted sum of the loss,

$$\mathcal{L}(u_\theta) = \frac{1}{n_{\text{int}}} \sum_{i=1}^{n_{\text{int}}} \|\mathcal{D}u_\theta(x_i) - f(x_i)\|_2^2 + \frac{1}{n_{\text{bdy}}} \sum_{j=1}^{n_{\text{bdy}}} \|u_\theta(z_j) - u^*(z_j)\|_2^2$$

where $x_i \sim \mathcal{U}([0, 1]^d)$ and where $z_j \sim \mathcal{U}(\partial[0, 1]^d)$ for $i = 1, \ldots, 10^5$ and $j = 1, \ldots, 5 \cdot 10^4$ both i.i.d. We observe that the splat equation converges very quickly and to a highly accurate solution, even when limited to use orders of magnitude fewer parameters the KAN and MLP models.

## 5 CONCLUSION

We have shown that splat models are highly effective in solving a variety of representative low dimensional modeling problems. We believe that these results are extremely promising from a practical perspective. Furthermore, we have drawn a novel connection between 3D Gaussian Splatting and Wasserstein-Fisher-Rao gradient flows, which we hope will lead to many symbiotic interactions between the optimal transport, computer graphics, and statistics communities.

There is much future work to be done. First, as we discuss in Section 3.2, splat models are highly expressive and therefore susceptible to overfitting, warranting a larger scale computational study beyond the scope of our work. The splatting community has developed many heuristics for regularizing and pruning splat models, which we expect will be instrumental to large-scale splat regression modeling. As a second direction, we remark that the proposed model can be interpreted as a new *neural network layer*, opening the door to compositions of splat models ('deep splat networks') and/or integrations into existing deep architectures. We look forward to investigating these directions in followup work.

ACKNOWLEDGMENTS

The authors wish to thank Ziang Chen, Tudor Manole, Youssef Marzouk, and Max Simchowitz for many helpful and friendly discussions. This material is based upon work supported by the U.S. Department of Energy, Office of Science, Office of Advanced Scientific Computing Research, Department of Energy Computational Science Graduate Fellowship under Award Number(s) DE-SC0023112.

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

## A   Well-posedness, Regularity, and Universal Approximation

Here we consolidate the proofs that are related to understanding structural properties of splat measures $\mu \in \mathcal{P}(\mathbb{R}^p \times \mathrm{BW}_\rho(\mathbb{R}^d))$ and splat functions. This includes 2 on the well-posedness and differentiability of splat functions and also the proofs of two universal approximation theorems, Proposition 3 and Theorem 3. We restate these claims for reading convenience.

**Proposition 2** (Sufficient conditions for regularity). Let $\mu \in \mathcal{S}_{p,d}$ and suppose that the mother splat $\rho$ has $s \geq 0$ bounded derivatives. Then $f_\mu(x)$ also has $s$ bounded derivatives.

*Proof.* Since $\rho \in C_b^s(\mathbb{R}^d)$ and for any $A \in \mathbb{R}^{d \times d}$, $b \in \mathbb{R}^d$, $x \mapsto Ax + b$ is $C_b^\infty(\mathbb{R}^d)$, the function $(v,x) \mapsto v\,[(A(\cdot) + b)_\# \rho](x)$ has $s$ bounded and continuous derivatives in $x$ and is dominated by the $\nu$-integrable function $v \mapsto C_s v \|\partial_s \rho\|_\infty \sup_{v \in \mathrm{supp}(\nu)} \|A\|_{\mathrm{op}}^s$ where $C_s$ is a universal constant. Derivatives of $f_\mu(\cdot)$ are therefore given by differentiation under the integral. $\square$

Next, we show that under a wide range of $\rho$, splat models satisfy the conditions of Cybenko's theorem on universal approximation by two-layer neural networks. We restate the universal approximation theorem and check that its conditions hold.

**Theorem 2** (Definition 1 and Theorem 1, Cybenko (1989)). *We say that $\sigma$ is discriminatory if for any measure $\mu \in I_n \coloneqq [0,1]^n$,*

$$\int_{I_n} \sigma(y^T x + \theta)\, d\mu(x) = 0$$

*for all $y \in \mathbb{R}^n$ and $\theta \in \mathbb{R}$ implies that $\mu = 0$. If $\sigma$ is any continuous discriminatory function, then the finite sums of the form*

$$G(x) = \sum_{j=1}^N \alpha_j \sigma(y_j^T x + \theta_j)$$

*are dense in $C_b^0(I_n)$.*

**Corollary 1.** *Let $\Omega \subseteq \mathbb{R}^d$ compact and suppose $\rho$ is a continuous density with marginal $\rho_1(x_1) = \int_{\mathbb{R}^{d-1}} \rho(x_1, x_2 \ldots x_d)\, dx_2 \ldots dx_d$. Then for any $f \in C_b^0(\Omega; \mathbb{R})$ and $\epsilon > 0$, there exists a $k$-splat measure $\mu$ (with mother splat $\rho$) such that*

$$\sup_{x \in \Omega} |f(x) - f_\mu(x)| < \epsilon.$$

*Proof.* By scaling we may assume $\Omega \subseteq [0,1]^d$. We claim that $\rho_1$ is a discriminatory function and that the approximator $G(x)$ with $\sigma = \rho_1$ and with parameters $\{(\alpha_j, y_j, \theta_j) : j = 1 \ldots k\}$, which is guaranteed by Theorem 2 for some $k \gg 1$, can be realized as a splat model. It is easy to see the latter: to realize $G(x)$ as a splat, take $A_{v_j} = e_1 y_j^T$, $b_{v_j} = e_1 \theta_j$, and $v_j = \alpha_j$.

Let $\mu$ be a nonzero measure on $[0,1]^d$ and let $s \in \mathrm{supp}(\mu)$. Assume for contradiction that for all $y \in \mathbb{R}^d, \theta \in \mathbb{R}$,

$$0 = \int_{[0,1]^d} \rho_1(y^T x + \theta)\, \mu(dx) \geq t\, P_{x \sim \mu}(\rho_1(y^T x + \theta) > t).$$

Thus $\mu(B_{y,\theta}(t)) = 0$ for all sets $B_{y,\theta}(t) \coloneqq \{x \in [0,1]^d : \rho_1(y^T x + \theta) > t\}$ and all $t > 0$. By continuity, the set $\rho_1^{-1}((t, \infty))$ is open, so that by affine rescaling we can construct from the basis $\{B_{y,\theta}(t) : y \in \mathbb{R}^d, t, \theta \in \mathbb{R}\}$ the entire Borel $\sigma$-algebra on $[0,1]^d$, which would imply $\mu = 0$. $\square$

Finally, we give a more constructive proof of a universal approximation theorem on sets $\Omega \subset \mathbb{R}^d$ whose uniform measure $\mathcal{U}(\Omega)$ has a finite *Poincaré constant*. This constant is the smallest $C_\Omega > 0$ for which any $f \in H^1(\Omega)$ satisfies the inequality

$$\mathrm{var}_{X \sim \mathcal{U}(\Omega)}(f(X)) \leq C \|\nabla f\|_{\mathcal{L}^2(\Omega)}^2.$$

This is satisfied by most 'nice' sets, namely whenever $\Omega$ is an open connected set with a Lipschitz boundary (Evans, 2010, §5.8 Theorem 1).

**Theorem 3.** *Let $\Omega \subseteq \mathbb{R}^d$ compact and suppose $f \in C_b^1(\Omega; \mathbb{R}^p)$. Suppose further that the measure $\omega = \mathrm{Unif}(\Omega)$ has finite Poincaré constant $C_\Omega < \infty$. For $\epsilon > 0$ there exists a $k$-splat measure $\mu$ at most $k \leq \frac{C_\Omega}{2} \epsilon^{-2(d+2)} \left( L_f B_\rho \epsilon + B_f L_\rho \right)^2$ splats and for which*

$$\sup_{x \in \Omega} \| f(x) - f_\mu(x) \| < (1 + L_f M_\rho) \epsilon.$$

*Where the constants are:*

$$
\begin{aligned}
B_\rho &= \| \rho \|_\infty & B_f &= \| f \|_\infty \\
L_\rho &= \sup_{x,y \in \mathbb{R}^d} \frac{|\rho(x) - \rho(y)|}{\| x - y \|_2} & L_f &= \sup_{x,y \in \Omega} \frac{\| f(x) - f(y) \|_2}{\| x - y \|_2} \\
M_\rho &= \mathbb{E}_{z \sim \rho}[\| z \|_2]
\end{aligned}
$$

*Proof.* Set $f^\epsilon(x) = \mathbb{E}_{z \sim \rho}[f(x + \epsilon z)]$, then

$$\| f(x) - f^\epsilon(x) \|_2 \leq \int \| f(x) - f(x + \epsilon z) \|_2 \, \rho(dz) \leq L_f M_\rho \epsilon$$

We now show that, for $k \gg 1$ also to be determined, there exist a $k$-splat measure $\mu$ so that

$$\sup_{x \in \Omega} \| f_\mu(x) - f^\epsilon(x) \|_2 \leq \epsilon.$$

Take random $x_1, \ldots, x_k \sim \omega^{\otimes k}$ and take $\nu = \frac{1}{k} \sum_{i=1}^k \delta_{f(x_i)}$, $k(v, x) = \epsilon^{-d} \rho(x/\epsilon)$, and $\mu(dv, x) dx = \nu(dv) k(v, x)$. Then we have

$$
P \left( x_1 \ldots x_k \mapsto \sup_{x \in \Omega} \| f_\mu(x) - f^\epsilon(x) \|_2 > \epsilon \right) \leq \frac{\mathrm{Var} \left( x_1 \ldots x_k \mapsto \sup_{x \in \Omega} \| f_\mu(x) - f^\epsilon(x) \|_2 \right)}{\epsilon^2}
$$

$$
\leq \frac{1}{\epsilon^2} \sum_{i=1}^k \mathbb{E}[\mathrm{Var}_i(x_1 \ldots x_k \mapsto \sup_{x \in \Omega} \| f_\mu(x) - f^\epsilon(x) \|_2)]
$$

$$
= \frac{1}{\epsilon^2} \sum_{i=1}^k \mathbb{E}[\mathrm{Var}_i(x_1 \ldots x_k \mapsto \sup_{\substack{x \in \Omega \\ r \in S^{p-1}}} \langle r, f_\mu(x) - f^\epsilon(x) \rangle)]
$$

where $\mathrm{Var}_i(f(x_1, \ldots, x_k))$ is the variance of $f(\cdot)$ holding $x_1, \ldots, x_{i-1}, x_{i+1}, \ldots, x_k$ fixed and resampling $x_i \sim \omega$ independently. In advance of applying Poincaré's inequality, invoke the envelope theorem to see,

$$
\nabla_{x_i} \left\{ x_1 \ldots x_k \mapsto \sup_{\substack{x \in \Omega \\ r \in S^{p-1}}} \langle r, f_\mu(x) - f^\epsilon(x) \rangle \right\} = (r^*)^T D_{x_i} \left\{ \frac{1}{k} \sum_{i=1}^k f(x_i) \epsilon^{-d} \rho((x - x_i)/\epsilon) \right\}
$$

$$
= \frac{1}{k \epsilon^d} (r^*)^T D f(x_i) \rho((x - x_i)/\epsilon)
$$

$$
- \frac{1}{k \epsilon^{d+1}} \langle r^*, f(x_i) \rangle (\nabla \rho)((x - x_i)/\epsilon)
$$

where $x^*, r^*$ attain the supremum. Its norm is bounded by

$$\| \ldots \| \leq \frac{1}{k \epsilon^{d+1}} \left( L_f B_\rho \epsilon + B_f L_\rho \right)$$

where $\| f \|_\infty := \sup_{x \in \Omega} \| f(x) \|_2$, and $C_\rho := \sup_{x,y \in \mathbb{R}^d} |\rho(x) - \rho(y)| / \| x - y \|_2$, so that by Poincaré inequality,

$$P(x_1, \ldots, x_k \mapsto \sup_{x \in \Omega} | f_\mu(x) - f^\epsilon(x) | > \epsilon) \leq \frac{C_\Omega}{\epsilon^{2(d+2)} k} \left( L_f B_\rho \epsilon + B_f L_\rho \right)^2.$$

Consequently, for $k = \frac{1}{2C_\Omega} \epsilon^{-2(d+2)} (L_f B_\rho \epsilon + B_f L_\rho)^2$, it occurs with probability at least one half (over $x_1, \ldots, x_k \sim \omega^{\otimes k}$) that $\sup_{x \in \Omega} \|f_\mu(x) - f^\epsilon(x)\|_2 \leq \epsilon$. By the triangle inequality

$$\sup_{x \in \Omega} \|f_\mu(x) - f(x)\|_2 \leq \sup_{x \in \Omega} \|f_\mu(x) - f^\epsilon(x)\|_2 + \|f^\epsilon(x) - f(x)\|_2 \leq (1 + L_f M_\rho)\epsilon.$$

$\square$

**Theorem 4** (Uniform approximation lower bound). *. Let $\rho$ be an $L_\rho$-lipschitz, $B_\rho$-bounded mother splat density. Fix $k \geq 1$ and $h > 0$, and set*

$$\mathcal{S}_h := \left\{ f := \sum_{i=1}^k v_i (A_i(\cdot) + b_i)_{\#}\rho \ : \ (v_i, A_i, b_i) \in \Gamma, \ i = 1 \ldots k \right\}$$

$$\mathcal{F} := \left\{ f : [0,1]^d \to \mathbb{R} : f(0) = 0, \ f \text{ is 1-Lipschitz} \right\}$$

*where $\Gamma$ contains all splat parameters $(v, A, b) \in \mathbb{R} \times \mathbb{R}^{d \times d} \times \mathbb{R}^d$ satisfying the bounds $|v| \leq B_v$, $\sigma_{min}(A) \geq h$, $\|b\|_2 \leq 1$, and where $\sigma_{min}(A)$ is the smallest singular value of $A$. If it holds that,*

$$\sup_{f \in \mathcal{F}} \inf_{\hat{f} \in \mathcal{S}_h} \|f - \hat{f}\|_\infty \leq \epsilon \tag{5}$$

*then (up to logarithmic factors) $\epsilon^{-d} \lesssim kd^2$ when $h, \epsilon$ are sufficiently small.*

*Proof.* We compare the metric entropies of $\mathcal{F}$ and $\mathcal{S}_h$. Let $\mathcal{P}(\epsilon, \mathcal{F})$, $\mathcal{N}(\epsilon, \mathcal{S}_h)$ be the packing and covering numbers of $\mathcal{F}$, $\mathcal{S}_h$ respectively, in the supremum norm on $[0,1]^d$. It is well known that $\log \mathcal{P}(\epsilon, \mathcal{F}) \gtrsim \epsilon^{-d}$ (Wainwright, 2019, Chapter 5). By assumption of equation 2 we also have $\log \mathcal{P}(\epsilon, \mathcal{F}) \leq \log \mathcal{N}(\epsilon/2, \mathcal{S}_h)$.

For any $\mu \in \Gamma^{\otimes k}$, the associated $f_\mu \in \mathcal{S}_h$ is given by

$$f_\mu(x) = \sum_{(v,A,b) \in \mu} v\,(A(\cdot) + b)_{\#}\rho(x) = \sum_{(v,A,b) \in \mu} v\,|\det(A^{-1})|\,\rho(A^{-1}(x-b)).$$

It can be rewritten in the form $f_\mu(x) = \sum_{i=1}^k u_i \rho(W_i(x - b_i))$ for some parameters $\{(u_i, W_i, b_i)\}_{i=1}^k$ with $\|u_i\|_2 \leq B_v h^{-d}$, $\|W_i\|_{op} \leq h^{-1}$, and $\|b_i\|_2 \leq 1$. Call $\bar{\Gamma} \subset \mathbb{R} \times \mathbb{R}^{d \times d} \times \mathbb{R}^d$ the set of all $(u, W, b)$ satisfying these bounds and call $\bar{\mathcal{S}}_h$ the functions,

$$\bar{\mathcal{S}}_h := \left\{ g(\cdot) := \sum_{i=1}^k u_i \rho(W_i^{-1}(\cdot - b_i)) \ : \ (u_i, W_i, b_i) \in \bar{\Gamma}, \ i = 1, \ldots, k \right\}.$$

Thus $\mathcal{S}_h \subseteq \bar{\mathcal{S}}_h$ and it is enough to bound $\log \mathcal{N}(\epsilon, \bar{\mathcal{S}}_h)$. As before, we denote by $g_{\bar\mu}$ the element of $\bar{\mathcal{S}}_h$ with parameters $\bar\mu \in \bar{\Gamma}^{\otimes k}$. For $\nu, \nu' \in \bar{\Gamma}^{\otimes k}$, $\nu = \{(u_i, W_i, b_i)\}_{i=1}^k$, $\nu' = \{(u_i', W_i', b_i')\}_{i=1}^k$, we have

$$\|g_\nu - g_{\nu'}\|_\infty \leq \sup_{x \in \mathbb{R}^d} \sum_{i=1}^k |u\,\rho(W(x-b)) - u'\,\rho(W'(x-b'))|$$

$$\leq \sup_{x \in \mathbb{R}^d} \sum_{i=1}^k |(u-u')\rho(W(x-b)) + u'(\rho(W'(x-b')) - \rho(W(x-b)))|$$

$$\leq \sup_{x \in \mathbb{R}^d} \sum_{i=1}^k B_\rho |u_i - u_i'| + B_v L_\rho h^{-d} \|W(x-b) - W'(x-b')\|_2.$$

For any $x \in [0,1]^d$,

$$\|W(x-b) - W'(x-b')\|_2 \leq \|(W-W')(x-b)\|_2 + \|W'(b-b')\|_2$$
$$\leq 2\sqrt{d}\|W - W'\|_{op} + h^{-1}\|b - b'\|_2.$$

Thus

$$\|g_\nu - g_{\nu'}\|_\infty \leq \sum_{i=1}^k B_\rho |u_i - u_i'| + 2h^{-d}\sqrt{d}B_v L_\rho \|W_i - W_i'\|_{op} + h^{-(d+1)}B_v L_\rho \|b_i - b_i'\|_2.$$

By tensorization, we can crudely bound

$$
\begin{aligned}
\log \mathcal{N}(\epsilon, \bar{\mathcal{S}}_h) \leq k \Bigg( & \log \mathcal{N}\left(\frac{\epsilon}{k B_\rho B_v h^{-d}}, \ |\cdot|, \ [0,1]\right) \\
& + \log \mathcal{N}\left(\frac{\epsilon}{2k\sqrt{d}B_v L_\rho h^{-d}}, \ \|\cdot\|_{op}, \ B_{op}\left(1, \mathbb{R}^{d\times d}\right)\right) \\
& + \log \mathcal{N}\left(\frac{\epsilon}{k B_v L_\rho h^{-(d+1)}}, \ \|\cdot\|_2, \ [0,1]^d\right) \Bigg) \\
\lesssim k \Bigg( & 1 \vee \log\left(\epsilon^{-1} k B_\rho B_v h^{-d}\right) + d^2 \cdot 1 \vee \log\left(\epsilon^{-1} k\sqrt{d} B_v L_\rho h^{-d}\right) \\
& + d \cdot 1 \vee \log\left(\epsilon^{-1}\sqrt{d}B_v h^{-(d+1)} L_\rho\right) \Bigg) \\
\lesssim k d^2 \cdot & 1 \vee \log\left(\epsilon^{-1} k\sqrt{d} h^{-(d+1)} B_v B_\rho L_\rho\right).
\end{aligned}
$$

Here, $B_{op}(1, \mathbb{R}^{d\times d})$ is the operator norm ball of radius 1 in $\mathbb{R}^{d\times d}$. Putting these parts together, we conclude that the minimax rate can only hold when

$$
\epsilon^{-d} \lesssim k d^2 \cdot 1 \vee \log\left(2\epsilon^{-1} k\sqrt{d} h^{-(d+1)} B_v B_\rho L_\rho\right).
$$

$\square$

## B  Gradient flows on measure spaces

We provide in this section some of the mathematical background on Wasserstein-Fisher-Rao calculus that is required to state our main results. We then give the proofs of Theorem 1, Proposition 1, and some helpful calculation rules for computing gradients with respect to splat measures.

### B.1  Review of Wasserstein and Fisher-Rao calculus

First, we define the Hellinger and the 'static' Wasserstein distances.

**Definition 3** (Wasserstein Distance (Villani, 2003)). Let $\mathcal{M}$ be a Riemannian manifold (endowed with the canonical volume measure) and let $\mu, \nu \in \mathcal{P}_p(\mathcal{M})$ be measures with finite $p$-th moments. The $p$-*Wasserstein* distance is,

$$
W_p^p(\mu, \nu) = \inf_{\pi \in \Gamma(\mu, \nu)} \int d_{\mathcal{M}}(X, Y)^p \, \pi(dX, dY)
$$

where $d_{\mathcal{M}}(\cdot, \cdot)$ is the geodesic distance on $\mathcal{M}$ and $\Gamma(\mu, \nu)$ is the set of all joint couplings whose marginals are $\pi_X = \mu$, $\pi_Y = \nu$. We denote by $\mathcal{W}_p(\mathcal{M})$ the space $\mathcal{P}_p(\mathcal{M})$ endowed with the $W_p$ metric.

**Definition 4** (Hellinger Distance (Amari, 1983)). Let $\mathcal{M}$ be as in 3 and let $\mu, \nu \in \mathcal{P}(\mathcal{M})$ be measures with finite second moments. Setting $\lambda = \frac{1}{2}(\mu + \nu)$, the Hellinger distance is

$$
H^2(\mu, \nu) = \int \left(\sqrt{\frac{d\mu}{d\lambda}(x)} - \sqrt{\frac{d\nu}{d\lambda}(x)}\right)^2 \lambda(dx)
$$

where $\frac{d\mu}{d\lambda}(x)$ is the Radon-Nikodym derivative. Here, $\lambda$ can be replaced by any measure $\lambda' \gg \mu, \nu$. We denote by $\mathcal{H}(\mathcal{M})$ the space $\mathcal{P}(\mathcal{M})$ endowed with the $H$ metric.

For our purposes we only need to consider rather two rather simple manifolds: the Euclidean space (with the canonical $\|\cdot\|_2$ metric) and the Bures-Wasserstein space $\mathrm{BW}_\rho(\mathbb{R}^d)$. We postpone the proof of Proposition 1, where we formally define the Bures-Wasserstein manifold, and proceed with the review. For further reference and for proofs of the background theorems stated here, see wonderful

reference (Santambrogio, 2015) as well as the more contemporary (Chewi et al., 2025a), which emphasizes a statistical perspective.

In their seminal 1998 paper, Jordan, Kinderlehrer, and Otto (Jordan et al., 1998) introduced an iterative 'minimizing movements' scheme as a variational approach to solving the Fokker-Plank equation in fluid dynamics, leading them (largely by accident) to discover the geometric perspective on the so-called 'static' Wasserstein distance. Shortly afterwards, Benamou & Brenier (2000) developed the following elegant formulation.

**Theorem 5** (Benamou–Brenier Benamou & Brenier (2000)). *Given two probability measures $\mu_0, \mu_1 \in \mathcal{W}_2(\mathcal{M})$, it holds that*

$$W_2^2(\mu_0, \mu_1) \tag{6}$$

$$= \inf_{\tilde{v}_t} \left\{ \int_0^1 \mathbb{E}_{X \sim \mu_t} \|\tilde{v}_t(X)\|^2 \, dt \, : \, \partial_t \mu_t + \mathrm{div}\,(\mu_t \tilde{v}_t) = 0, \mu_{t=0} = \mu_0, \mu_{t=1} = \mu_1 \right\}. \tag{7}$$

*where $\|\tilde{v}_t(x)\|^2 := \langle \tilde{v}_t(x), \tilde{v}_t(x) \rangle_x$ is the tangent space norm at $x \in \mathcal{M}$. Moreover, the optimizer $\{v_t\}$ induces a curve $\{\mu_t\}$ with the following properties.*

1. *At time $t = 0$, $v_0(x) = T_{\mu_0 \to \mu_1}(x) - x$ where $T_{\mu_0 \to \mu_1}(x)$ is the optimal transportation map from $\mu_0$ to $\mu_1$.*

2. *$(v_t)_{t \geq 0}$ has zero acceleration, in the sense that $v_t(X_t(x)) = T_{\mu_0 \to \mu_1}(x) - x$ is constant in time.*

3. *For times $t > 0$, $\mu_t = \mathsf{Law}\,[X + t(T_{\mu_0 \to \mu_1}(X) - X)]$ with $X \sim \mu_0$, and the joint law of $(X, X + t(T_{\mu_0 \to \mu_1}(X) - X))$ is the optimal coupling of $(\mu_0, \mu_t)$.*

The integrand of equation 6 is called the 'kinetic energy' of the ensemble $\mu_t$ and the minimizing curves $(\mu_t)_{t \in [0,1]}$ are the geodesics of $\mathcal{W}_2(\mathcal{M})$. One can show by analyzing the Euler-Lagrange equations of equation 6 that the tangent vectors $(v_t)_{t \in [0,1]}$ have the form $v_t(x) = \nabla \phi_t(x)$. Conversely, for any curve of measures $(\nu_t)_{t \in [0,1]}$ that is *absolutely continuous* in the following sense, there exists a $\phi_t$ whose gradient weakly solves the transport equation $\partial_t \nu_t = -\mathrm{div}_{\mathcal{M}}(\nu_t \nabla \phi_t)$. Evidently $\phi_t$ itself is the solution of a Poisson equation with positive semidefinite coefficents, which already enjoys strong existence, uniqueness, and regularity theorems Evans (2010).

**Definition 5** (Absolute continuity). Let $(\mu_t)_{t \in [0,1]} \in \mathcal{W}_2(\mathcal{M})$ be a curve of measures. The *metric derivative* $|\dot{\mu}|(t)$ is defined,

$$|\dot{\mu}|(t) = \lim_{h \to 0} \frac{W_2(\mu_{t+h}, \mu_h)}{h}$$

and $\mu_t$ is *absolutely continuous* (in $W_2$ metric) at $t \in [0,1]$ if $|\dot{\mu}|(t) < \infty$.

**Theorem 6.** *If $(\mu_t)_{t \in [0,1]}$ is absolutely continuous, then there exists a function $\phi_t : \mathcal{M} \to \mathbb{R}$ which weakly solves*

$$\partial_t \mu_t + \mathrm{div}_{\mathcal{M}}(\mu_t \nabla \phi_t) = 0$$

*and moreover for which $\nabla \phi_t \in \mathcal{L}^2(\mu_t)$ for every $t \in [0,1]$.*

The tangent spaces of $T_\mu \mathcal{W}_2(\mathcal{M})$ are defined in the usual way as the linear span of the tangent vector of any absolutely continuous curve passing through $\mu_t$. It is customary to identify the tangent vector $\partial_t \mu_t = -\mathrm{div}_{\mathcal{M}}(\mu_t \nabla \phi_t)$ with its drift function $\nabla \phi_t$, since the metric $\langle \cdot, \cdot \rangle_\mu$ has a simple expression in terms of the drifts.

**Theorem 7.** *The space $\mathcal{W}_2(\mathcal{M})$ is a Riemannian manifold with tangent space structure*

$$T_\mu \mathcal{W}_2(\mathcal{M}) = \overline{\{v \in \mathcal{L}^2(\mu) : \exists \phi : C_c^\infty(\mathcal{M}) \to \mathbb{R}, \nabla \phi = v\}}_{\mathcal{L}^2(\mu)}$$

*with inner product $\langle u, v \rangle_\mu = \mathbb{E}_{X \sim \mu} \langle u(X), v(X) \rangle_X$, where $\langle \cdot, \cdot \rangle_X$ is the metric on $T_X \mathcal{M}$.*

Finally, the Wasserstein gradient $\mathbb{W} \mathcal{F}(\mu)$ is just the represener of $\xi \mapsto \partial_{\epsilon=0} \mathcal{F}(\mu + \epsilon \xi)$ in $T_\mu \mathcal{W}_2(\mathcal{M})$.

**Definition 6** (Wasserstein Gradient). Let $\mathcal{F} : \mathcal{M} \to \mathbb{R}$, the Wasserstein gradient at $\mu$ is (when it exists) the function $\overline{\mathbb{W}}\mathcal{F}(\mu) \in T_\mu \mathcal{W}_2(\mathcal{M})$ such that for every $\xi \in T_\mu \mathcal{W}_2(\mathcal{M})$,

$$\partial_{\epsilon=0}\mathcal{F}(\mu_\epsilon) = \langle \delta\mathcal{F}, \xi \rangle_\mu = \int \langle \delta\mathcal{F}(X), \xi(X) \rangle_X \, \mu(dX).$$

where $(\mu_t)_{t \in (-\delta, \delta)}$ is any curve with tangent vector $\xi$ at $\mu_0 = \mu$.

We turn our attention to the Fisher-Rao geometry, which is more straightforward to explain. A standard reference for this material is the book Amari (2016). The 'admissible' tangent vectors in this geometry are those belonging to the absolutely continuous curves,

**Definition 7** (Fisher-Rao absolute continuity). A curve $(\mu_t)_{t \in [0,1]}$ is *absolutely continuous* at $t \in [0, 1]$ if its Hellinger metric derivative is finite:

$$|\dot{\mu}|(t) := \lim_{\epsilon \to 0} \frac{H(\mu_{t+\epsilon}, \mu_t)}{\epsilon} < \infty.$$

**Theorem 8.** *If $(\mu_t)_{t \in [0,1]} \in \mathcal{H}(\mathcal{W})$ is absolutely continuous at $t \in [0, 1]$, then there exists $\alpha_t : \mathcal{M} \to \mathbb{R}$ such that $\partial_t \mu_t = \alpha_t \mu_t$, and where $\mathbb{E}_{X \sim \mu_t}[\alpha_t(X)] = 0$.*

The Fisher-Rao geometry is essentially based on the fact that for any probability density $\mu(x) : \mathcal{M} \to \mathbb{R}$, the square root density $\sqrt{\mu}(x)$ is an element of the unit sphere $S(\mathcal{L}^2(\mathcal{M})) := \{v \in \mathcal{L}^2(\mathcal{M}) : \|v\| = 1\}$, since it has $\int (\sqrt{\mu}(x))^2 \, dx = 1$ trivially. We endow it with the pullback metric on $\mathcal{P}(\mathcal{M})$ induced by the map $\iota : \mu \mapsto \sqrt{\mu} \in S(\mathcal{L}^2(\mathcal{M}))$. What that means concretely is:

- The geodesic connecting $\mu_0, \mu_1$ is the spherical linear interpolant connecting $\sqrt{\mu_0}, \sqrt{\mu_1}$:

$$\sqrt{\mu_t} = \left( \frac{\sin((1-t)\theta)}{\sin(\theta)} \right) \sqrt{\mu_0} + \left( \frac{\sin(t\theta)}{\sin(\theta)} \right) \sqrt{\mu_1}, \qquad \cos(\theta) = \langle \sqrt{\mu_0}, \sqrt{\mu_1} \rangle$$

- Each tangent space $T_\mu \mathcal{H}(\mathcal{M})$ is by definition isomorphic to $T_{\sqrt{\mu}} \mathcal{H}(\mathcal{M})$. The isomorphism map is the pushforward or 'differential' $d\iota$, which is given by

$$d\iota(\xi) := \partial_\epsilon \, \iota(\mu + \epsilon\xi) \Big|_{\epsilon=0} = \frac{\xi}{2\sqrt{\mu}}$$

and so the inner product on $T_\mu \mathcal{H}(\mathcal{M})$ is

$$\langle \xi, \psi \rangle_\mu = \langle d\iota(\xi), d\iota(\psi) \rangle_{\sqrt{\mu}} = \frac{1}{4} \int \frac{\xi\psi}{\mu}(x) \, dx.$$

- By absolute continuity we may write $\xi = \alpha\mu$, $\psi = \beta\mu$ for densities $\alpha, \beta : \mathcal{M} \to \mathbb{R}$, and the Fisher-Rao metric can be rewritten as

$$\langle \alpha\mu, \beta\mu \rangle = \frac{1}{4} \int \alpha(x)\beta(x)\mu(dx).$$

It is customary to identify the tangent vectors with their densities and to write

$$T_\mu \mathcal{H}(\mathcal{M}) = \overline{\{\alpha \in C_c^\infty(\mathcal{M}) : \mathbb{E}_\mu[\alpha(X)] = 0\}}_{\mathcal{L}^2(\mu)}$$

with metric $\langle \cdot, \cdot \rangle_\mu = \frac{1}{4}\langle \cdot, \cdot \rangle_{\mathcal{L}^2_\mu(\mathcal{M})}$.

- The Fisher-Rao gradient of a functional $\mathcal{F}$ is (when it exists) the function $\nabla^{\mathsf{FR}}\mathcal{F} \in T_\mu \mathcal{H}(\mathcal{M})$ that satisfies, for $\alpha \in T_\mu \mathcal{H}(\mathcal{M})$,

$$\partial_\epsilon \mathcal{F}(\mu + \epsilon\alpha\mu) \Big|_{\epsilon=0} = \langle \nabla^{\mathsf{FR}}\mathcal{F}(\mu), \alpha \rangle_\mu.$$

Finally, the Wasserstein-Fisher-Rao geometry is the one whose tangent spaces are direct sums of the Wasserstein and the Fisher-Rao tangent spaces. In other words, Wasserstein-Fisher-Rao tangent vectors have the form

$$\partial_t \mu_t + \mathrm{div}_{\mathcal{M}}(\mu_t \nabla\phi_t) = \alpha_t \mu_t, \qquad \nabla\phi_t \in T_\mu \mathcal{W}(\mathcal{M}), \ \alpha_t \in T_\mu \mathcal{H}(\mathcal{M}).$$

and as usual we identify each tangent vector with its coefficients $(\nabla\phi_t, \alpha_t)$. The Wasserstein-Fisher-Rao metric is

$$\langle(\nabla\phi_t, \alpha_t), (\nabla\psi_t, \beta_t)\rangle_\mu = \frac{1}{4}\int \alpha_t(x)\beta_t(x)\,\mu(dx) + \int \langle\nabla\phi_t(x), \nabla\psi_t(x)\rangle\,\mu(dx),$$

and its tangent spaces are

$$T_\mu\mathcal{W}(\mathcal{M}) \oplus T_\mu\mathcal{H}(\mathcal{M}) = \overline{\{(a, v) : \exists\,\alpha, \phi \in C_c^\infty(\mathcal{M}) \to \mathbb{R},\ v = \nabla\phi\}}^{\mathcal{L}^2(\mu)},$$

and the Wasserstein-Fisher-Rao gradient of $\mathcal{F}$ is $\nabla^{\mathsf{WFR}}\mathcal{F}(\mu) = (\mathbb{W}\mathcal{F}(\mu), \nabla^{\mathsf{FR}}\mathcal{F}(\mu))$.

## B.2 PROOFS OF PROPOSITION 1 AND THEOREM 1

The proof of Proposition 1 follows exactly the same steps as the case $\rho = \mathcal{N}(0, I_d)$.

**Proposition 1** (Splats are a generalized Bures-Wasserstein manifold) **.** Let $\rho \in \mathcal{P}_{ac}(\mathbb{R}^d)$ be a centered isotropic mother splat. We denote the set of all splats as,

$$\mathsf{BW}_\rho(\mathbb{R}^d) := \left\{(A(\cdot) + b)_{\#}\,\rho : A \in \mathbb{R}^{d\times d}, b \in \mathbb{R}^d\right\}.$$

Then $\mathsf{BW}_\rho(\mathbb{R}^d)$ is a geodesically convex subset of $\mathcal{W}_2(\mathbb{R}^d)$, and on this space the Wasserstein metric reduces to the *Bures-Wasserstein metric* (Modin, 2016; Bhatia et al., 2019),

$$W_2^2(\rho_{A,b}, \rho_{R,s}) = \|b - s\|_2 + \|A\|_F^2 + \|R\|_F^2 - 2\|A^T R\|_* \qquad A, R \in \mathbb{R}^{d\times d} \quad b, s \in \mathbb{R}^d$$

where $\|\cdot\|_F$ is the Frobenius norm and $\|\cdot\|_*$ is the nuclear norm.

*Proof.* Fix $\rho_{A_0, b_0} = (A_0(\cdot) + b_0)_{\#}\rho$ and $\rho_{A_1, b_1} = (A_1(\cdot) + b_1)_{\#}$. By Brenier's uniqueness theorem (Chewi et al., 2025a, Theorem 1.16), the map

$$T(x) = b_1 + A_0^{-1}(A_0 A_1 A_1^T A_0)^{1/2} A_0^{-1}(x - b_0)$$

is the optimal transport map $T_{\#}\rho_{A_0, b_0} = \rho_{A_1, b_1}$ as it is the gradient of a convex function. It follows that the geodesic connecting $\rho_0$ to $\rho_1$ is $\rho_t = ((1-t)\mathsf{id} + tT)_{\#}\rho_0 \in \mathsf{BW}_\rho(\mathbb{R}^d)$. As $\rho$ is centered and isotropic, the distance $W_2(\rho_0, \rho_1)$ is

$$W_2(\rho_0, \rho_1) = \mathbb{E}_\rho\left[\|(A_0 X + b_0) - T(A_0 X + b_0)\|^2\right]$$

$$= \mathbb{E}_\rho\left[\|b_0 - b_1 + (A_0 - A_0^{-1}(A_0 A_1 A_1^T A_0)^{1/2})X\|\right]$$

$$= \|b_0 - b_1\|^2 + \|A_0 - A_0^{-1}(A_0 A_1 A_1^T A_0)^{1/2}\Sigma_X^{1/2}\|_F^2$$

where $\Sigma_X = \mathrm{Cov}_\rho(X) = I_d$. Thus $W_2(\rho_0, \rho_1) = W_2(\mathcal{N}(b_0, A_0 A_0^T), \mathcal{N}(b_1, A_1 A_1^T))$ is the Bures-Wasserstein metric Bhatia et al. (2019). $\square$

We now proceed to the proof of Theorem 1. We develop these calculations at a formal level – the reader who wishes to understand an entirely rigorous proof may compare to Ambrosio et al. (2008, Chapter 11).

**Theorem 1** (Wasserstein-Fisher-Rao gradient of $\mu \mapsto \mathcal{F}(f_\mu)$) **.** Let $\mu \in \mathcal{S}_{p,d} := \mathcal{P}(\mathbb{R}^p \times \mathsf{BW}_\rho(\mathbb{R}^d))$ and let $\mathcal{F} : \mathcal{L}^2(\mathbb{R}^d; \mathbb{R}^p) \to \mathbb{R}$ be a functional. Assume that $\rho$ has a sub-exponential density. Then the Fisher-Rao gradient is given (if it exists) by the formula,

$$\nabla_\mu^{\mathsf{FR}}\mathcal{F}(f_\mu)(v, A, b) = \mathbb{E}_{X\sim\rho_{A,b}}[\langle\delta\mathcal{F}(X), v\rangle] - \mathbb{E}_{v,A,b\sim\mu}[\mathbb{E}_{X\sim\rho_{A,b}}[\langle\delta\mathcal{F}(X), v\rangle].$$

and the Wasserstein gradient is given (if it exists) by the formula,

$$\mathbb{W}_v\mathcal{F}(f_\mu)(v, A, b) = \mathbb{E}_{X\sim\rho_{A,b}}[\delta\mathcal{F}(X)]$$

$$\mathbb{W}_A\mathcal{F}(f_\mu)(v, A, b) = -\mathbb{E}_{X\sim\rho_{A,b}}\left[\langle\delta\mathcal{F}(X), v\rangle\left(I_d + \nabla_x\log\rho_{A,b}(X)(X - b)^T\right)A^{-T}\right]$$

$$\mathbb{W}_b\mathcal{F}(f_\mu)(v, A, b) = -\mathbb{E}_{X\sim\rho_{A,b}}[\langle\delta\mathcal{F}(X), v\rangle\nabla_x\log\rho_{A,b}(X)]$$

where the argument of $\delta\mathcal{F}(\cdot) = \delta\mathcal{F}[f](\cdot)$ was suppressed.

*Proof.* We begin by calculating the Fisher-Rao gradient. Along the way, we develop a 'chain rule,' which may aid in future calculations. Specifically we view $\mu \mapsto f_\mu$ as a mapping between manifolds, $f_{(\cdot)} : \mathcal{H}(\mathcal{S}_{p,d}) \to \mathcal{L}^2(\mathbb{R}^d; \mathbb{R}^p)$, and we calculate its *differential* $df_\mu : T_\mu \mathcal{H}(\mathcal{S}_{p,d}) \to \mathcal{L}^2(\mathbb{R}^d; \mathbb{R}^p)$. For $\alpha \in T_\mu \mathcal{H}(\mathcal{S}_{p,d})$,

$$d^{FR} f_\mu[\alpha](x) = \partial_\epsilon f_{\mu+\epsilon\alpha\mu} \big|_{\epsilon=0} = \int_{\mathcal{S}_{p,d}} v\, \alpha(v, A, b) \rho_{A,b}(x)\, \mu(dv, dA, db).$$

Invoking the definition of the first variation, our chain rule takes the form

$$\partial_\epsilon \mathcal{F}(f_{\mu+\epsilon\alpha}) = \langle \delta\mathcal{F}[f_\mu], df_\mu[\alpha] \rangle_{\mathcal{L}^2(\mathbb{R}^d;\mathbb{R}^p)}$$

$$= \iint \langle \delta\mathcal{F}[f_\mu](x), v \rangle\, \alpha(v, A, b)\, \rho_{A,b}(x)\, dx\, \mu(dv, dA, db)$$

$$= \int \alpha(v, A, b) \left( \mathbb{E}_{X \sim \rho_{A,b}}[\langle \delta\mathcal{F}[f_\mu](X), v \rangle] \right)\, \mu(dv, dA, db).$$

We identify the Fisher-Rao gradient by matching the integrand to the definition.

Computing the Wasserstein gradient follows largely the same steps. One checks the following auxiliary calculations.

$$\nabla_A \left\{ \rho\left(A^{-1}(x-b)\right) |\det A|^{-1} \right\} = -|\det A|^{-1} A^{-T} (\nabla\rho)(A^{-1}(x-b))(x-b)^T A^{-T}$$

$$- \rho\left(A^{-1}(x-b)\right) |\det A|^{-1} A^{-T}$$

$$= -\rho_{A,b}(x) \left( I + \nabla_x \log \rho_{A,b}(x)(x-b)^T \right) A^{-T}$$

$$\nabla_b \left\{ \rho\left(A^{-1}(x-b)\right) |\det A|^{-1} \right\} = -(\nabla\rho)(A^{-1}(x-b))|\det A|^{-1}$$

$$= -\nabla_x \log \rho_{A,b}(x)\, \rho_{A,b}(x).$$

Now fix $u \in T_\mu \mathcal{W}_2(\mathcal{S}_{p,d})$. We calculate the differential $d^W f_\mu[u]$ as,

$$d^W f_\mu[u](x) = -\int_{\mathcal{S}_{p,d}} v\rho_{A,b}(x)\, \mathrm{div}_{\mathcal{S}_{p,d}}(u\mu)(dv, dA, db)$$

$$= \int \langle \nabla_{v,A,b} \{v\rho_{A,b}(x)\}, u(v, A, b) \rangle\, \mu(dv, dA, db)$$

$$= \int u_v(v, A, b)\rho_{A,b}(x)\, \mu(dv, dA, db)$$

$$+ \int v \langle \nabla_A \{\rho_{A,b}\}(x), u_A(v, A, b) \rangle\, \mu(dv, dA, db)$$

$$+ \int v \langle \nabla_b \{\rho_{A,b}\}(x), u_b(v, A, b) \rangle\, \mu(dv, dA, db)$$

where $u = (u_v, u_A, u_b)$ are the coordinate function representations of $u$. Thus, setting $\xi = -\mathrm{div}_{\mathcal{S}_{p,d}}(u\mu)$,

$$\partial_\epsilon \mathcal{F}(f_{\mu+\epsilon\xi}) = \langle \delta\mathcal{F}[f_\mu], df_\mu[u] \rangle_{\mathcal{L}^2(\mathbb{R}^d;\mathbb{R}^p)}$$

$$= \iint \langle \delta\mathcal{F}[f_\mu](x), u_v(v, A, b) \rangle \rho_{A,b}(x)\, dx\, \mu(dv, dA, db)$$

$$+ \iint \langle \delta\mathcal{F}[f_\mu](x), v \rangle \langle \nabla_A \{\rho_{A,b}\}(x), u_A(v, A, b) \rangle\, dx\, \mu(dv, dA, db)$$

$$+ \iint \langle \delta\mathcal{F}[f_\mu](x), v \rangle \langle \nabla_b \{\rho_{A,b}\}(x), u_b(v, A, b) \rangle\, dx\, \mu(dv, dA, db).$$

From this we see,

$$\mathbb{W}_v \mathcal{F}(f_\mu)(v, A, b) = \int \delta\mathcal{F}[f_\mu](x)\, \rho_{A,b}(x)\, dx$$

$$\mathbb{W}_A \mathcal{F}(f_\mu)(v, A, b) = -\int \langle \delta\mathcal{F}[f_\mu](x), v \rangle (I + \nabla_x \log \rho_{A,b}(x)(x-b)^T) A^{-T}\, \rho_{A,b}(x)\, dx$$

$$\mathbb{W}_b \mathcal{F}(f_\mu)(v, A, b) = -\int \langle \delta\mathcal{F}[f_\mu](x), v \rangle \nabla_x \log \rho_{A,b}(x)\, \rho_{A,b}(x)\, dx.$$

$\square$

There is an interesting relationship between the geometry of $\mathcal{S}_{p,d}$ and that of the space $\mathcal{P}(\mathbb{R}^p \times \mathbb{R}^d)$. Intuitively, splats in $\mathsf{BW}_\rho(\mathbb{R}^d)$ are like 'smoothed particles,' this relationship bears out concretely when calculating the Wasserstein and Fisher-Rao gradients in $\mathcal{P}(\mathbb{R}^p \times \mathbb{R}^d)$.

**Theorem 9** (Wasserstein-Fisher-Rao gradient of $\mu \mapsto \mathcal{F}(f_\mu)$ for $\mu \in \mathcal{P}(\mathbb{R}^p \times \mathbb{R}^d)$)**.** *Let $\mu \in \mathcal{P}(\mathbb{R}^p \times \mathbb{R}^d)$ and let $\mathcal{F} : \mathcal{L}^2(\mathbb{R}^d; \mathbb{R}^p) \to \mathbb{R}$ be a functional. Assume that $\rho$ has a sub-exponential density. Then the Fisher-Rao gradient is given (if it exists) by the formula,*

$$\nabla_\mu^{\mathsf{FR}}\mathcal{F}(f_\mu)(v, x) = \langle \delta\mathcal{F}(x), v \rangle - \mathbb{E}_{X,V \sim \mu}[\langle \delta\mathcal{F}(X), V \rangle].$$

*and the Wasserstein gradient is given (if it exists) by the formula,*

$$\mathbb{W}_v \mathcal{F}(f_\mu)(v, x) = \delta\mathcal{F}(x)$$

$$\mathbb{W}_x \mathcal{F}(f_\mu)(v, x) = -v^T D_x \mathcal{F}(x)$$

*where the argument of $\delta\mathcal{F}(\cdot) = \delta\mathcal{F}[f](\cdot)$ was suppressed.*

Comparing to the Wasserstein-Fisher-Rao gradients in Theorem 1, we see that the gradients have the same form as Theorem 9, but they are averaged with respect to the splat density. The $\mathbb{W}_b \mathcal{F}$ has the same relationship as can be seen by integrating by parts,

$$\mathbb{W}_b \mathcal{F}(f_\mu)(v, A, b) = -\mathbb{E}_{X \sim \rho_{A,b}}[\langle v, \delta\mathcal{F}(X) \rangle \nabla_x \log \rho_{A,b}(x)] = \mathbb{E}_{X \sim \rho_{A,b}}[v^T D_x \delta\mathcal{F}(X)].$$

These formulas follow from the following expressions for $d^W f_\mu$, $d^{FR} f_\mu$, which can be derived by the same arguments as for Theorem 1.

$$\langle \delta\mathcal{F}[f_\mu], d^W f_\mu(u) \rangle := \int \langle \delta\mathcal{F}[f_\mu] \otimes v^T D_x \delta\mathcal{F}(f_\mu), u(v, x) \rangle \, \mu(dv, dx)$$

$$\langle \delta\mathcal{F}[f_\mu], d^{FR} f_\mu(\alpha) \rangle := \int \langle \delta\mathcal{F}[f_\mu], v \rangle \alpha(v, x) \, \mu(dv, dx).$$

## C EXPERIMENTAL DETAILS AND ADDITIONAL EXPERIMENTS

### C.1 WALL CLOCK RUNTIMES

We report the average wall clock runtime to train each of the models in Figures2 run on a 2020 Macbook Pro M1. We also include each model's parameter count and final MSE for comparison.

| Model Type | Parameter count | Wall Time (s) | Final MSE |
|---|---|---|---|
| KAN | 521 | 24.2853 | 0.2129 |
| KAN | 5201 | 121.9557 | 0.1404 |
| KAN | 7861 | 930.1017 | 0.0045 |
| KAN | 15601 | 358.9623 | 0.0728 |
| KAN | 20801 | 529.0655 | 0.1781 |
| MLP | 801 | 7.6694 | 0.2266 |
| MLP | 2001 | 8.7870 | 0.2165 |
| MLP | 4001 | 10.5255 | 0.2052 |
| MLP | 41001 | 12.1382 | 0.0390 |
| MLP | 252501 | 22.5081 | 0.0087 |
| SPLAT | 60 | 51.9213 | 0.0575 |
| SPLAT | 300 | 104.6052 | 0.0016 |
| SPLAT | 600 | 149.7740 | 0.0005 |
| SPLAT | 1200 | 243.9303 | 0.0006 |
| SPLAT | 1800 | 295.8554 | 0.0005 |
| SPLAT | 2400 | 378.7603 | 0.0008 |

Although they have significantly fewer parameters, training splat models has a more expensive wall clock time. We use autodifferentiation to compute gradients with respect to the splat model parameters, and the appearance of $A^{-1}$ and $\det A^{-1}$ in equation equation 1 are particularly in our non-optimized implementation. We expect that a faster implementation with hardcoded gradient formulas will be significantly faster. Further, as can be seen in Figures 2 and 3, the splat models converge in far fewer iterations than KAN and MLP, so the reported run times include the time required to take many unnecessary training steps for the splat models.

## C.2 ADDITIONAL APPROXIMATION EXPERIMENTS

The following plots show versions of our function approximation experiment in two different settings. The first shows the same experiment, but for fitting a sawtooth target function. The second shows the same experiment, but with $\{b_i\}_{i=1}^k$ initialized as a $k$-point chebyshev grid.

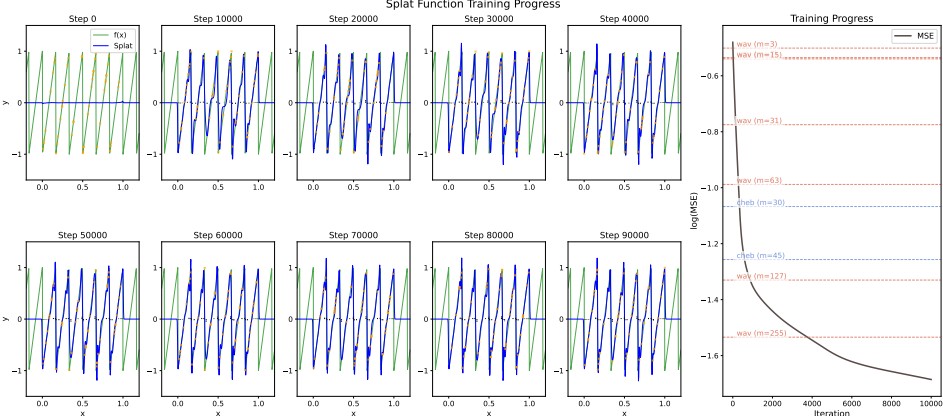

Figure 4: Fitting a sawtooth function in the setting of 1. This is a much harder function to fit with interpolation methods. Perhaps surprisingly, splats outperforms the Haar wavelet decomposition, which can exactly fit vertical discontinuities.

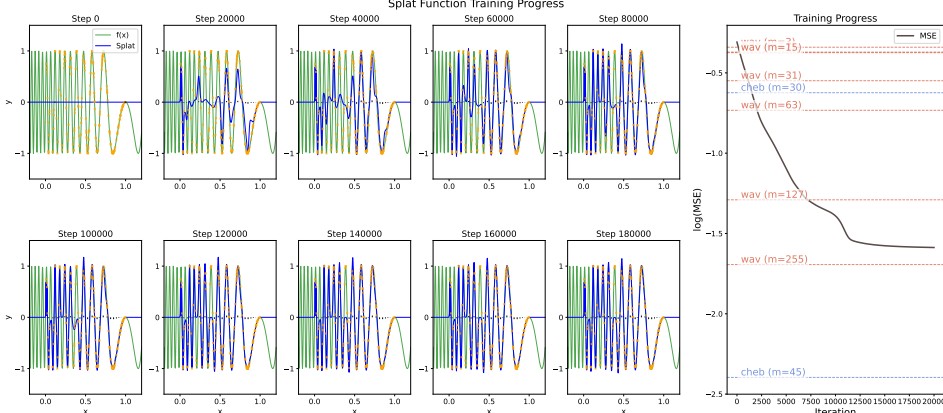

Figure 5: In the setting of 1, we test the effect of Chebyshev initialization, which is slightly less performant.

