## A  WELL-POSEDNESS, REGULARITY, AND UNIVERSAL APPROXIMATION

Here we consolidate the proofs that are related to understanding structural properties of splat measures $\mu \in \mathcal{P}(\mathbb{R}^p \times \mathrm{BW}_\rho(\mathbb{R}^d))$ and splat functions. This includes Propositions 1, 2 on the well-posedness and differentiability of splat functions. We also provide the proof of the two universal approximation theorems, Proposition 3 and Theorem 3. We restate these claims for reading convenience.

We dedicate the rest of the section to proving Theorem 1, which we split into a few steps.

**Proposition 1** (Heterogeneous Mixtures). Suppose that $\mu \in \mathcal{P}(\mathbb{R}^p \times \mathbb{R}^d)$ is of the form $\mu(dv, dx) = \mu(dv, x)\, dx$ and where $\mathbb{E}_{(v,x)\sim\mu}[\|v\|_s^s] < \infty$, for $s \geq 1$. Then the function $f_\mu(x) := \mathbb{E}_{v\sim\mu(\cdot,x)}[v]$ belongs to $\mathcal{L}^s(\mathbb{R}^d; \mathbb{R}^p)$, and furthermore, there exists $\nu \in \mathcal{P}(\mathbb{R}^p)$ (the 'mixture components') and a density $k(v, x) > 0$ (the 'heterogeneous mixture weights') so that

$$\mu(dv, x) = \nu(dv)\, k(v, x) \qquad f_\mu(x) = \mathbb{E}_{v\sim\nu}[vk(v, x)].$$

*Proof.* If $\mathbb{E}_{(v,x)\sim\mu}[\|v\|_s^s]$ then by Jensen's inequality,

$$\|f_\mu\|_{\mathcal{L}^s(\mathbb{R}^d;\mathbb{R}^p)}^s = \int \left\| \int v\, \mu(dv, x) \right\|_s^s dx \leq \iint \|v\|_s^s\, \mu(dv, x)\, dx < \infty$$

so $f_\mu \in \mathcal{L}^2(\mathbb{R}^d; \mathbb{R}^p)$. Now take $\nu$ to be the $v$-marginal $((v, x) \mapsto v)_{\#}\mu$. By the disintegration theorem (Ambrosio et al., 2008) there exists a family of measures $k_v \in \mathcal{P}(\mathbb{R}^d)$ so that $\mu(dv, x)dx = k_v(dx)\nu(dv)$. We must check that each $k_v$ has a density, and for this it is sufficient to check that for any $\phi \in \mathcal{L}_b^1(\mathbb{R}^p)$ and any sequence $\psi_k \in \mathcal{L}^1(\mathbb{R}^d)$ that converges $\phi_k \overset{k\to\infty}{\longrightarrow} 0$ in $\mathcal{L}^1$,

$$\int \phi(v) \left( \int \psi_k(x)\, k_v(dx) \right) \nu(dv) = \iint \phi(v)\psi_k(x)\mu(dv, dx)$$

$$\leq \sup_{v\in\mathrm{supp}(\nu)} \phi(v) \cdot \|\psi_k\|_{\mathcal{L}^1(\mathbb{R}^d)}$$

$$\overset{k\to\infty}{\longrightarrow} 0.$$

by Hölder's inequality. We have shown that for $\nu$-almost every $v$, any sequence converging to zero in $\mathcal{L}^1(\mathbb{R}^d)$ also converges to zero in $\mathcal{L}^1(\mathbb{R}^d; k_v)$. The proof follows by identifying $k(v, x)$ as the density of $k_v(dx)$. $\square$

Next we prove the sufficient conditions on $\mu$ for $f_\mu(\cdot)$ to be continuous and/or differentiable.

**Proposition 2** (Sufficient conditions for regularity). Let $\mu \in \mathcal{P}(\mathbb{R}^d \times \mathbb{R}^d)$.

1. The map $f_\mu(\cdot)$ has uniform modulus of continuity

$$\omega(\epsilon) = \sup \left\{ \left| \mathbb{E}_{\mu(\cdot,x)}[v] - \mathbb{E}_{\mu(\cdot,y)}[v] \right| : \|x - y\| \leq \epsilon \right\}$$
$$\leq \sup \left\{ W_1(\mu(\cdot, x), \mu(\cdot, y)) : \|x - y\| \leq \epsilon \right\}.$$

   In particular, $f_\mu(\cdot)$ is absolutely continuous whenever $(\mu(\cdot, x))_{x\in\mathbb{R}^d}$ is a $W_1$-absolutely continuous measure-valued process on $\mathbb{R}^d$.

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

$$\sup_{f \in \mathcal{F}} \inf_{\hat{f} \in \mathcal{S}_{1,d}^{(k)}} \|f - \hat{f}\|_\infty \gtrsim k^{-1/d}$$

where the inequality holds up to universal constants.

*Proof.* We invoke some well known facts about the metric entropy of $\mathcal{F}$ and of the parametric class $\mathcal{S}_{1,d}^{(k)}$ Wainwright (2019). The metric entropy (in uniform norm) of $\mathcal{F}$ scales as $H(\epsilon, \mathcal{F}, \|\cdot\|_\infty) \sim \epsilon^{-d}$, whereas by parameter counting, one has $H(\epsilon, \mathcal{S}_k, \|\cdot\|_\infty) \sim k \log(\epsilon^{-1})$. Uniform approximation is therefore impossible unless $H(\epsilon, \mathcal{F}, \|\cdot\|_\infty) \lesssim H(\epsilon, \mathcal{S}_k, \|\cdot\|_\infty)$, which forces the rate $k \gtrsim \epsilon^{-d}$ when $\epsilon < 1$. $\