# OpenReview forum: "Splat Regression Models"
_ICLR.cc/2026/Conference — ICLR 2026 Poster_

### Official Review · Reviewer_MULa · 2025-10-30

**Soundness:** 3
**Presentation:** 3
**Contribution:** 3
**Rating:** 6
**Confidence:** 3

**Summary:**

The paper presents the idea of a splat model which from my understanding is an extension of RBF's of Kernel neural networks. The paper focus on theory and shows that the method has a potential to do well for certain class of problems.

**Strengths:**

The paper is strong theoretically although I **did not verify the theoretical results**.

**Weaknesses:**

The paper presents very little numerical evidence. This is not necessarily bad and one has to weigh the novelty in the theory if the paper is to be accepted.

**Questions:**

1. None of the experiments is done on "standard" data sets.
The method seems to be doing better than other methods on simple problems but can you show a single problem that is not a toy problem. I am not looking for SOTA, just to see that you are in the same ball park.

2. The paper is dense with theory. What are the major new results here. Some look rather trivial

---

### Official Review · Reviewer_D7Ct · 2025-10-31

**Soundness:** 2
**Presentation:** 3
**Contribution:** 2
**Rating:** 4
**Confidence:** 3

**Summary:**

This paper introduces Splat Regression Models (SRMs), a new class of function approximators based on mixtures of anisotropic, heterogeneous bump functions (“splats”). The authors frame SRMs as a generalization of classical nonparametric regression and show that 3D Gaussian Splatting emerges as a special case. They develop a theoretical foundation using Wasserstein–Fisher–Rao (WFR) gradient flows for optimization over the space of mixing measures and demonstrate strong empirical performance on low-dimensional regression and physics-informed tasks, outperforming MLPs and Kolmogorov–Arnold Networks (KANs).

**Strengths:**

* This work provides a unified theoretical framework that recovers 3D Gaussian Splatting as a special case of a broader class of function approximators
* It introduces a rigorous optimization scheme based on Wasserstein–Fisher–Rao (WFR) gradient flows over the space of mixing measures. This gives a principled justification for heuristic practices in Gaussian Splatting
* Each “splat” acts as a learnable, anisotropic, localized basis function—effectively learning an adaptive interpolation grid. This offers better interpretability than black-box MLPs and more flexibility than fixed bases
*  SRMs are shown to be universal approximators, with a quantitative bound on the number of splats needed to approximate Lipschitz functions.

**Weaknesses:**

* All experiments are in 1D or 2D. The theoretical approximation bound scales as $k \le \varepsilon^{-2(d+2)}$, indicating a severe curse of dimensionality. This limits applicability to problems beyond very low-dimensional scientific modeling. Nevertheless, for low dimensional modeling problems It will be helpful to provide comparisons with the same baseline (MLP, KAN)  in terms of memory and running time for both training and inference. It is also important the authors to clarify what is their definition of low dimensionality .
* Experiments use synthetic or toy PDEs (e.g., Allen–Cahn on a unit square). There is no validation on real-world data from physics, biology, or engineering, which significantly weakens any claims about practical relevance.
* SRMs are closely related to adaptive RBF networks, mixture density networks, and mesh-free methods, but the authors do not adequately contrast their approach with these methods.
* The authors acknowledge that SRMs are highly expressive and prone to overfitting but they don't provide concrete regularization strategies in the main experiments, which would be critical for deployment in real-world problems.

**Questions:**

The practical scope of this work is currently narrow, and the broader impact hinges on addressing scalability, regularization, and real-world validation.

---

### Official Review · Reviewer_ojFa · 2025-11-02

**Soundness:** 4
**Presentation:** 3
**Contribution:** 4
**Rating:** 8
**Confidence:** 4

**Summary:**

The paper presents a new class of function approximators parameterized by a mixture of scaled and shifted bump functions that they call splats. The authors introduce these as generalizations of the popular Gaussian Splatting approach. They show that such functions act as universal approximators and present an optimization approach for fitting such models on data through Wasserstein-Fisher-Rao gradient flows. The paper presents examples of applications including physics informed training and empirical results for multiscale training and a physics solver.

**Strengths:**

1. The presented framework is a principled generalization of Gaussian Splatting style approaches allowing for clear interpretable function families as approximators.

2. For physics-based problems and similar low-dimensional spaces, the inductive bias represented by `splats' is a natural match. \

3. The approach seems to be more parameter efficient than MLPs and KANs with smoother and faster convergence characteristics, at least for the simple, representative problems presented here.

4. The WFR lens for splatting reveals interesting future directions into principled regularization approaches compared to the heuristics used currently.

**Weaknesses:**

1. The current empirical experiments are quite limited with simple settings and baselines. I suggest adding a few more baselines like Random-fourier-features + MLP,  and more complex settings (perhaps 3D regression, or higher order PDEs).

2. While Gaussian splatting (NVS) is mentioned theoretically, there are no empirical results validating this. Specifically, even a simple experiment validating theorem 1 under the novel view synthesis setting.

3. I believe the inner matrix transformation map $T(.)$ should be $A_0 A_1 A^T_1 A_0^T$ for it to be PD in order for results from Bhatia, 2019 to hold. Please feel free to correct me if my understanding is wrong.

**Questions:**

See weaknesses above.

---

### Official Review · Reviewer_5ihC · 2025-11-06

**Soundness:** 3
**Presentation:** 3
**Contribution:** 3
**Rating:** 6
**Confidence:** 3

**Summary:**

Paper proposes new function approximators, named as Splat Regression Models (SRMs). It is built as a mixture of finite number of localized splats (e.g. anisotropic Gaussians), where they learn positions, shapes, and weights during the training. SRMs resemble 2 layer neural network architecture with different activation function. It is framed in a measure theoretic way, and splat parameter updates are analyzed/derived carefully with Wasserstein-Fisher-Rao (WFR) gradients. Theory also provides regularity inheritance of mother splat and universal approximation bounds for this proposed "new layer". SRMs achieve lower error rates with fewer parameters than MLPs/KANs on low-dimensional regressions, inverse problems, and PDE tasks. Paper further recovers common 3D Gaussian splatting framework as a special case of SRM.

**Strengths:**

1. Paper provides clean and important "neural network component/layer" that is built upon Gaussian splats. Idea is well principled under measure-theoretic formulation and WFR gradient flow framework.
2. Theoretical results on regularity and WFR gradient derivations are solid and plausible. Geometric perspective is also well-crafted.
3. Paper is well-written as it narrates the topic in a very intuitive way and also provides important theoretical results in a rigorous way.
4. Experiments on low-dimensions/PDEs outperform main baselines with smaller number of parameters, which shows the expressivity of SRMs.
5. It also frames 3D gaussian splatting as a special case in a principled way.

**Weaknesses:**

1. The bound on $k$ being $\epsilon^{-2(d+2)}$ is likely loose. Authors also acknowledge that they use comparably smaller $k$'s in practice, which suggests that better theory could be developed to improve this bound.
2. For low-dimensional tasks, more ablations could be better. Given that model has this overfitting effect, it might worth to discuss also add some experiments with regularization tricks as discussed to be future work in the paper. This has been acknowledged but there is not any principled regularizer provided. Moreover, the baselines for these tasks could be extended to tuned classical kernels or more PDE regimes.
3. There are not experiments on high-dimensional tasks. Given that the bound is loose, the higher dimension scaling of method is very intriguing. Authors propose deep splat networks by stacking proposed layers, but it would be beneficial to have some initial set of experiments on this side.

**Questions:**

1. Can you provide wall-clock time, FLOPs, and/or GPU memory comparisons? SRMs has less number of parameters, but it is interesting to see if optimization process or gradient calculation part has some overhead compared to baselines.
2. Can you add some initial experiments (i) with regularization techniques; (ii) against better tuned kernels, RBF networks; (iii) on high-dimensions or deep splat idea?
3. I wonder about the analysis on sensitivity of $k$ and initialization strategies of $(A, b, v)$. Can you provide study on that showing different combinations to clarify why the current choices are best?
4. For PDE regression tasks, why there are not many baselines as there are plenty of them there with physics-informed ideas and neural operators?
5. Have authors conducted any experiments on 3D gaussian splats? It is proven to be special case theoretically, but is SRM pipeline scalable to this with WFR gradients?

---

### Meta-Review · Area_Chair_GMPJ · 2026-01-04

**Summary:**

This paper presents a new class of function approximators parameterized by a mixture of scaled and shifted bump functions. Moreover, it introduces a rigorous optimization scheme based on Wasserstein-Fisher-Rao gradient flows over the space of mixing measures. All reviewers gave this paper positive scores. From my own reading, I think it merits acceptance in ICLR.

**Reviewer Concerns:**

Most concerns are addressed by the authors, and they gave a detailed response to these rebuttals.

**Reviewer Scores:**

Reviewer D7Ct would increase his/her score if he/she had been able to participate fully in the discussion.

---

### Decision · Program_Chairs · 2026-01-26

Accept (Poster)